# Research Progress and Production Status of Edible Insects as Food in China

**DOI:** 10.3390/foods13131986

**Published:** 2024-06-24

**Authors:** Boxuan Xie, Yuxuan Zhu, Xiaoyi Chu, Sabin Saurav Pokharel, Lei Qian, Fajun Chen

**Affiliations:** 1Department of Entomology, College of Plant Protection, Nanjing Agricultural University, Nanjing 210095, China; xieboxuan@stu.njau.edu.cn (B.X.); zhuyuxuan@stu.njau.edu.cn (Y.Z.); chuxiaoyi@stu.njau.edu.cn (X.C.); pokharelsabin93@gmail.com (S.S.P.); 2Institute of Leisure Agriculture, Jiangsu Academy of Agricultural Sciences, Nanjing 210014, China

**Keywords:** edible insects, nutrient components, functional values, artificial rearing technology, alternate food sources, food insecurity, sustainable agriculture

## Abstract

Based on the background of the exacerbating food shortage in the world, it is particularly important to diversify food resources in every possible direction. Among the choices available, edible insects have become an important alternative source of animal food with their high nutritional and functional (pharmacological) values, partially replacing normally consumed animal and livestock protein food sources. The utilization of edible insects has been an ancient custom since the dawn of civilization, attributed to their rich nutrition, alternate protein source, medicinal values, and presence of diverse secondary metabolites and alkaloids. This review provides an introduction to three key aspects of edible insects as food: freshness, long-term preservation, and medicinal value. It also provides details on the food source and products of edible insect species, their detailed nutritional composition and medicinal values, and their potential in producing alternative protein sources. Additionally, the review also encompasses rearing and producing technologies, resource utilization, and industrial development in China. Simultaneously, the problems and challenges faced in the artificial rearing and production development of edible insects, the production advantages over traditional livestock, and the farming evaluation and prospects of edible insects, as well as the lack of specific legislation on edible insects in China, are discussed. This review will be helpful in scientific knowledge propagation regarding edible insects for the public, guiding consumers to establish a diverse perception of sustainable agriculture and food sources in the world that has, as yet, been thwarted by food insecurity. Moreover, though edible insects could potentially serve as part of a commercial and industrial agri-enterprise that could generate a huge income, artificial rearing technology and edible insect product manufacturing and processing have not received sufficient attention from the government on a policy level, thereby leaving an open space for extensive research on edible insects as an alternate food source as well as an examination of the industrial prospects of edible insect products.

## 1. Introduction

The consumption of edible insects has been an ancient practice among a significant portion of the global population, particularly in Latin America, Africa, and Asia [1]. The scientific community has been fascinated with entomophagy across different global cultures, and there is a rich historical background of insects serving as a regular dietary choice, a means of survival, a source of medicinal benefits, a part of ritualistic practices, and as accidental meals [2,3]. Recently, “The Food and Agriculture Organization” (FAO) of the United Nations has been promoting the concept of insects as a source of food. The FAO put forth the idea that insects can play a crucial role in addressing the current global issue of food scarcity, considering the limitations of current food production and agricultural systems to meet the current needs of a growing global population [4]. Additionally, the “2017 Central Rural Work Conference of China (CRWCC)” also highlighted the current case of people’s food preferences and the seeking of alternate food sources. The conference emphasized the need to transform the traditional food concept by integrating a wider array of protein sources derived from cultivated land, grassland, forests and oceans, as well as plants, animals and microorganisms. Edible insects fit perfectly with the notions strained by FAO and the CRWCC, where they act as a more environmentally sustainable, affordable, and feasible alternative protein source that can alleviate the global demand for food supplies. The proverb that “Insect resources are the last piece of cake left to mankind by God” emphasizes the importance of including edible insects in our daily diet supply [5].

Edible insects refer to those insects whose bodies or products can be used as food and medicine [6]. Several scientific studies have indicated that humans have been consuming insects in their diet since the dawn of civilization [7]. The consumption of edible insects is very common across the globe, including all continents when food resources are scarce. According to research conducted by the International Centre of Insects Physiology and Ecology (ICIPE), it was discovered that Africa is home to 470 species of edible insects [8]. Consumption of edible crickets was common in African nations like Tanzania, Zimbabwe and Botswana, while the sub-Saharan Africans preferred consuming caterpillars (Lepidoptera) [1,9]. Conversely, Europeans preferred to eat locusts, beetles and ants, and Sweden was the heart of edible insect-consuming culture in Europe [10,11]. Central Americans created a renowned insect cuisine from moths called “moth cake”, while Japanese people have prepared culinary dishes from edible insects, including locusts, bee larvae and pupa, and silkworm pupa. People from Pakistan and Egypt consumed decorative beetles, while Thai and Arabs preferred “fried grasshopper” [10,12]. Additionally, other famous edible insect cuisines include Mexican “braised agave aphids” and “ant food” [12,13]. In China, early records of eating bumblebee larvae or pupae are mentioned in the famous ancient books “Erya, the Rites of Zhou” and “The Book of Rites”. Nowadays, people in the north of China prefer consuming locusts, silkworm pupae, golden cicada and DanBean (i.e., larvae of *Clanis bilineata tingtauica*). Meanwhile, people living in Jiangsu and Zhejiang provinces preferred to consume silkworm pupae as an edible insect food [14]. Those living in the Fujian and Guangdong provinces liked to eat dragon lice, and the residents in the Taiwan region liked to eat crispy crickets, while the people in the western Hunan Province preferred fried wasp larvae [15]. The long history of insect consumption is attributed to their extremely high nutritional and unique medicinal values. There are many functional substances in edible insects, e.g., antimicrobial peptides, interferon, sex attractant hormones, cordycepin, active polysaccharides, microelement, chitin or chitosan, steroidal material and vitamin, lecithin, etc., which could provide multiple benefits for human beings, and these special functional substances could help people to enhance their immunity, inhibit tumors, regulate intestinal functions, relieve fatigue, protect against colds, improve sleeping, promote growth and development as well as reduce blood sugar and blood pressure [16]; edible insect are also believed to have anti oxidation effects. Edible insects are rich sources of proteins, fats, amino acids and carbohydrates, etc. For instance, the famous Chinese caterpillar fungus of *Cordyceps sinensis* has multi-purpose medicinal benefits, such as sedation, hemostasis, anticonvulsant and antihypertensive elements, and it also aids in improving myocardial ischemia [17]. Based on the nutritional and medicinal values of edible insects, there is still a great potential development prospect regarding edible insects in food resource utilization.

Edible insects are categorized as fresh food, long-term preservation food, and medicinal or functional health food based on the insect products prepared from larvae (nymphs), pupae, or adults. In order to meet the needs of consumers, some farmers or enterprises have carried out large-scale artificial farming of edible insects, and researchers are also actively exploring new technologies, methods, or processes to better develop edible insect farming to take full advantage of the nutritional and medicinal values of edible insects. In this review, our prime focus will be on exploring the species of edible insects and their edible products, farming and producing technology, nutrition and medicinal value, resource utilization and industry development, as well as their production advantages over traditional livestock, the potential in producing alternative protein sources, and the farming evaluation and the prospects of edible insects. The paper will also investigate the problems and challenges of edible insect consumption, as well as the lack of specific legislation on edible insects in China. The review also aims to break the conventional cliches surrounding edible insects as a food source and promote the production of edible insects and industry development as a normal commercial agri-enterprise.

## 2. Major Edible Insect Species and Their Resource Utilization in China

People from several countries across Asia, as well as in Africa, North America, and Europe, have exhibited the habit of eating insects [14]. In Europe, common edible insects include crickets and yellow mealworms. The European Food Safety Authority (EFSA) has granted approval for the sale and consumption of several species of edible insects as “novel food” for humans in European countries, including the domestic cricket (*Acheta domesticus*) and the yellow mealworm (*Tenebrio molitor*) [11,18]. These common edible insects are primarily manufactured into powders (e.g., cricket powder) and incorporated into foodstuffs (e.g., bread and cookies) to enhance their nutritional values. A review study indicated that bread containing cricket powder exhibits a highly variform composition of fatty acids and higher contents of protein and essential amino acids compared to the control breads made from pure wheat flour [19]. On the other hand, crickets are the mostly common accepted form of edible insect in North America [20].

The food made from edible insects (including larvae, nymphs, pupae, or adults) or using their products as the main raw material is called insect food [21]; the type directly eaten in the original form after simple processing and cooking is called original insect food [22]. To date, golden cicadas, DanBean (*Clanis bilineata tsingtauica* larvae), silkworm pupae, wasps, and locusts are the major edible insects in China, and their main nutrients and active ingredients are seen in Figure 1.

### 2.1. Golden Cicada

*Crytotympana pustulata* (Homoptera: Cicadidae), also known as golden cicada, is widely distributed in mainland China, northern India, Japan, and South Korea. The golden cicada is recognized by different names, including old turtle, pot, slit, black cicada, cicada turtle, and cicada monkey, and it undergoes incomplete metamorphosis and has the three distinct developmental stages of egg, nymph, and adult (seen in Figure 2). In nature, the lifespan of the golden cicada is approximately 3–5 years. After mating, female adults usually lay eggs in the tissues under the phloem of young branches (mostly annual branches) of trees, causing the dead branches to fall on the ground in autumn [23]. As people’s living standard continues to improve, the market demand for golden cicadas is also increasing. The increasing market demand for golden cicadas can be attributed to their pleasant taste, high nutritional value, and beneficial medicinal properties [24]. In recent years, the artificial rearing of golden cicadas has developed rapidly under controlled conditions (well-maintained temperature and humidity) from where the cicada eggs can hatch in 30–50 days as compared to the rigorous hatching that might take 6 months to one year [25]. After cicada eggs hatch into nymphs, the branches with cicada nymphs are buried under the roots of the host plants for 18 months, and the cicada nymphs emerge from the soil and climb on tree trunks, from where they are collected, packaged, and transported to the restaurants for public consumption. At present, the annual sales revenue of golden cicadas in China exceeds RMB 100 million, equivalent to 15 million US dollars [26].

#### 2.1.1. History and Status of Golden Cicada Consumption

Since ancient times, the custom of cicada consumption has existed in many places in the world, including China. The ancient Chinese literature “*Mao Shi Lu Shu Guang Yao*” mentioned big and yellow cicadas called ‘*Gai Tiao*’ which were consumed as edible insect food. These cicadas were caught, finely chopped, cooked, and savored by putting coriander leaves on the top. The highly revered ancient Chinese agricultural text “*Qimin Yaoshu*”, written by the Northern Wei Dynasty (368–534 AD) official Jia Sixie, mentioned three cooking methods of cicadas as insect food (i.e., roasting, steaming and boiling). In China, it has become customary to consume cicadas, but pinpointing the origin of this prevailing custom is an insurmountable task. The ‘Osmanthus Cicada’ dish served in Cantonese restaurants of the Guangdong Province, the cicada cuisine served at banquets in Tai’An of the Shandong Province, and a series of health food products branded as “Fairy Cicada” launched in the Henan Province, are highly preferred by the consumers [27]. Greek historian Herodotus (4th Century BC), often referred to as the “Father of History”, documented the practice of insect consumption among the ancient Greeks and Romans. In the past, Greeks and Romans would often harvest cicada nymphs from the soil and prepare pleasant cuisines [28]. Athenaeus (200 AD) mentioned cicada cuisines being served in Greek banquets to stimulate the appetite. The indigenous tribes of Malaysia and nearby archipelagoes of Southeast Asia also have the custom of consuming cicadas in their diet, but the difference is that they are likely to eat freshly caught cicadas, similar to the practices seen among the indigenous people of Australia and African countries [8,29]. In 1885, the British Entomologist Vincent M. Holt put forward the novel idea of including edible insects as a food source in his book “*Why Not Eat Insects?*” [30].

The golden cicada nymph has a unique taste and high nutritional value, thus making it a subtle savor as one of the best edible insects available. At present, the golden cicada has become an essential cuisine on the Chinese dining table. Just a mere plate of “Fried Golden Cicadas” has become a favorite delicacy of consumers because of its high-protein and low-fat nutritional value, with prices ranging from RMB 40 to 100 (equivalent to USD 0.6–14). In recent years, with the continuous improvement of people’s living standards, golden cicadas are frequently seen on the menus of grand hotels and big restaurants, sometimes even being served as canned or packaged delicacies where they are directly exported overseas [31]. With the continuous attention being given to green and healthy diets, it is easy to imagine that the green, natural, nutritious, and delicious high-protein food of edible insects will be favored by a large consumer base and thus have broad market prospects [29,30].

#### 2.1.2. Nutrient Composition and Medicinal Value of Golden Cicadas


*Nutrient Composition*


According to ancient Chinese pieces of literature, people of the Tang Dynasty highly preferred golden cicadas as part of their diet, and this high nutritional value diet was known as “Tang Priest’s Meat”. Delving into the individual nutrient composition, golden cicada constitute 72 g protein, 15 g fat, 1.8 g ash, and 1.36 mg vitamin B_1_ per 100 g body weight of golden cicada nymphs, while the vitamin B_1_ content gradually increases with the development of nymphs into adults, and the average content of vitamin B_1_ is found to be 2.88 mg/100 g body weight of golden cicadas [32]. On the other hand, golden cicadas are also rich in essential amino acids and able to supply 46.63% of the total amino acids required for the human body, making it an obvious alternative high-protein food source [33]. Some studies indicated that live golden cicadas constitute 58.58% protein, which is 3.5, 4.3, 3.8, 3, 4, and 6 times that of lean beef, lean pork, mutton, chicken, carp and eggs, respectively [33]. These scientific findings corroborate the fact that the golden cicada is a rare natural, high-protein, low-fat, and pollution-free alternative protein source [33]. In brief, the consumption of golden cicadas can have some positive effects on people’s health, but the public should also be careful when eating golden cicadas, as they contain abundant heterologous protein molecules which may cause allergic reactions for some people.


*Medicinal Value*


The prolific use of golden cicadas in traditional Chinese medicine has been described by several classic and scientific pieces of literature, e.g., “*Divine Farmer’s Classic of Materia Medica*” and “*Holy Prescriptions for Universal Relief*”. According to the former literature, salty and cold cicadas are used to treat children’s epileptic sleep terrors and soothe night crying, while the latter mentions the use of lightly fried cicadas along with seven pieces of dried scorpion (directly used), which, taken along with peppermint soup, can treat gastrointestinal diseases like bezoar and realgar as well as children’s eye twitches. The medicinal values of golden cicadas have been mentioned in the ancient traditional Chinese medicine literature [27]. According to “*Compendium of Materia Medica*”, autumn cicadas and *Fructus kochiae* (fruit of an annual potherb *Kochia scoparia*) are medium-well fried and used in a prepared medicine mix (5 g) of powder with wine to cure tetanus. The book “*Chinese Medicinal Materials*” further points out that cicadas have numerous pharmacological properties that enhance and strengthen vigor, slake thirst and produce saliva, protect lungs and benefit kidneys, resist bacteria, reduce blood pressure, treat baldness, and suppress cancer. The shell of cicadas (cicada slough/cicada skin) contains the main components of chitin and protein as well as adenosine triphosphatase, and cicada slough is sweet in taste and moist, also serving in an important Chinese medicine often utilized to target ailments associated with external wind heat, coughs, hoarseness, and sore throats via acupuncture around the meridian region of the lungs and liver; it is also used to treat rubella itching, red eyes, tetanus, children’s panic, and frequent night crying in traditional Chinese medicine [33]. Chitin derived from cicada slough has significant medicinal values, as it has the efficiency of strengthening the liver, reducing blood pressure, effective anti-coagulant properties, antibacterial properties, anti-aging properties, anti-diabetic, and anti-obesity properties, easing bowel movements and analgesic properties [33]. The chitin-based materials derived from cicadas have been widely used in biomedicine and tissue engineering, making them remarkably expensive in recent years. The future scope for the development and utilization of cicadas and their slough is very bright. Therefore, the golden cicada is not only a part of a traditional edible insect diet but also a common Chinese herbal medicine, and it can be said that the golden cicada as a whole insect is indeed a hidden treasure [31].

#### 2.1.3. Artificial Rearing Technology of Golden Cicada

The golden cicada inhabits a wide range of host trees, including more than 140 kinds of trees, but its most common host trees are poplar, willow, elm, and sycamore, as well as a variety of fruit trees (apple, pear, hawthorn and peach). Under normal circumstances, golden cicadas can grow well without special feeding requirements, paving the pathway for artificial rearing. The artificial rearing technology mainly includes the following steps, as described by Lu et al. (2019) [34]: (1) *Prepare egg branches.* It is appropriate to buy egg branches produced in local or adjacent areas, and it is best to buy white-ash egg branches. The number of tree branches with eggs laid by golden cicadas should be prepared according to the size and growth trend, and the number of egg branches can be appropriately increased with the growth of the tree age. Considering all together, each white wax egg branch contains 100–400 cicada eggs and at least 7500–12,000 egg branches per hectare of woodland. (2) *Egg branch preservation.* Golden cicada eggs can be preserved under natural conditions with controlled conditions. Temperature, humidity, and ventilation are the key artificial factors that should be maintained for egg branch preservation. Egg branches should be stored in a cool and well-ventilated place, and sun exposure should be avoided. Humidity is also very important for the preservation of golden cicada egg branches. Dry branches containing cicada eggs are prone to dying due to dryness, while extremely wet branches are prone to developing mold. Therefore, according to the specific situation of the golden cicada egg branches, fog water should be irregularly sprayed to adjust the humidity. Fog water spraying should be more frequent in autumn and spring and less frequent in winter so that the egg branches are maintained in a suitable humidity. When preserving the egg branches of golden cicadas indoors, the egg branches should be turned in a timely manner. Compared with the good ventilation conditions of cicada eggs under natural conditions, for branches with a large number of eggs, it is necessary to arrange them in rows, leaving a gap between them to allow for air circulation and to avoid overheating and molding of eggs. (3) *Cicada egg hatching.* Golden cicada eggs generally begin to hatch in late June and continue until mid−late August every year. When golden cicada eggs undergo the development phase, reddish eyes appear first, which gradually deepen into black, and ultimately the cicada eggs begin to gradually hatch as nymphs. When nymphs hatch, the preservation temperature of the egg branch should be maintained well, elevated up to 28 °C till 3 p.m., and the temperature increase is fully terminated at 6–7 p.m. until the temperature comes on par with the natural temperature. Before preparing for incubation, the egg branches are soaked in clean water and placed neatly on the shelf. The sand basin is placed under the shelf, maintaining the relative humidity of the sand manually. A plastic sheet with a diameter smaller than the circular hole of the sand basin is placed on the sand basin to undertake the newly hatched golden cicada nymphs.

### 2.2. DanBean

The larvae of the soybean hawk moth (*Clanis bilineata tingtauica*), also known as DanBean in China, belong to the family Sphingidae of Lepidoptera, which exclusively feeds on soybean leaves [35,36]. *C. bilineata tingtauica* undergoes complete metamorphosis and has four distinct developmental stages of egg, larva, pupa, and adult (seen in Figure 3). *C. bilineata tingtauica* is particularly common in most of the Chinese agricultural heartlands of the Shandong, Henan, and Jiangsu provinces. *C. bilineata tingtauica* adults are about 5 cm long and light green with a darker head and tail horns, and they have seven distinct pairs of white stripes on their dorsal and ventral sides from the first abdominal segment. The larvae of *C. bilineata tingtauica* (i.e., DanBean) are prolific leaf feeders, and their damage can be marked with the distinct holes in soybean leaves after their leaf penetration during feeding. In severe cases, these notorious pests damage entire fields of soybean plants, with no pods formation occurring in the soybeans after flowering. Although DanBean larvae harm soybean plants, they are a natural and organic food source with a delicious flavor and high palatability (as seen in Figure 1). Many farmers harvest these edible insects for medicinal purposes and also to fill the dietary requirements of the local people as an alternative protein source [35,36]. They are part of the most representative local-specific foods, especially in the nearby areas of Lianyungang City of the Jiangsu Province of China. The popular, unique, and delicious dish known as ‘Guanyun DanBean’, native to the Guanyun County, located in northern Jiangsu, has been mentioned by the national television network ‘China Central Television’ (CCTV) in the news reports “*Science Fair*” and “*Zhi Fu Jing*”.

#### 2.2.1. History and Current Situation of DanBean Consumption

The custom of DanBean consumption in China dates back to ancient times. The ancient Chinese scholar ‘Pu Songling’ wrote and described the correct way of DanBean consumption in his book ”*Classic of Silkworm Rearing*”. In Guanyun County, of the Jiangsu Province, the consumption of DanBean by the locals exceeded more than a dozen tons in July and August every year. As a result, the price of pakchoi (*Brassica rapa var. chinensis*), which is fried with DanBean, also rose sharply. Therefore, many people have discovered and explored ways to scale up pakchoi cultivation. The DanBean is very delicious, palatable, and pleasant, and it is preferred by the people as a culinary aesthetic. In the provinces which are known as China’s agri-heartlands, namely Guangdong, Jiangsu, Anhui, Shandong and Henan, the demand for DanBean is on the rise. In Guanyun County and Lianyungang City of the Jiangsu Province in particular, DanBean is famous as a local traditional food, where utilization of DanBean is more comprehensive, and they make a variety of dishes with their own traditional-styled cooking methods [37]. In recent years, entertaining guests using DanBean has been common in the Jiangsu and Shandong provinces [38].

With the improvement of people’s living standards, people’s demand for green and high-quality edible insect food is also increasing. At present, Guanyun County has become the largest DanBean food market in China, which promotes the industrialization development of DanBean utilization and also drives the rapid development of related enterprises. In 2019, the farming area for DanBean in Guanyun County of the Jiangsu Province was nearly 1334 hm^2^; the output reached 0.26 million tons and DanBean’s sales reached nearly RMB 1 billion [39]. It is prominent that the market potential of edible DanBean is huge and great. However, DanBean often appears in short supply in the current situation. Therefore, research on artificial rearing technology and commercial production of DanBean has high economic value and practical significance [36].

#### 2.2.2. Nutrition and Medicinal Value of DanBean


*Nutritional Value*


DanBean is also known as the elixir of life due to its medicinal properties, rich nutrition, and wide range of uses [40]. Since DanBean directly feeds on soybean leaves, it is believed to be a natural, pollution-free green, healthy and organic food with unique value in both the nutritional and medicinal aspects [41]. Compared with meat, fish, and plant proteins, the proteins of DanBean are more suitable for human nutrition needs, and the reasonable protein composition makes it an ideal high-quality protein resource for human beings [40]. In dry weight, the protein content of DanBean proteins is 63.2%, and it contains 18 types of amino acids [40]. The essential amino acids found in higher contents in DanBean ((methionine (2.39%), cystine (1.79%), alanine (2.38%), tyrosine (2.35%), tryptophan (2.75%)) are relatively high. Furthermore, other amino acids present in DanBean include isoleucine and lysine, accounting for 47.23% of the total amino acids, which is a moderate proportion for human development [40]. The proportion of essential amino acids in the total amino acids (52.84%) in DanBean is higher than that of eggs (50.94%), milk (47.46%) and soybeans (43.58%) [42]. The content of essential amino acids in DanBean adults is far more than the WHO recommended standard, and the crude protein content in DanBean is as high as 65.5% (fold dry weight), while the content of linolenic acid in essential amino acids is as high as 36.53%; additionally, semi-essential amino acids account for 9.70% of the total amino acids [43]. The crude fat content is as high as 23.68%, Go-GS fatty acids account for more than 99% of the total fatty acids, and unsaturated fatty acids are as high as 64.17% [42]. The chitin present in the epidermis of the oldest larvae is synthesized from trehalose in hemolymph, thereby making DanBean a vital edible insect with high trehalose content. In addition, the abundance of trace elements required by the human body and the high calcium, magnesium iron and vitamin E contents of the insect in question make DanBean a distinct edible insect [40].


*Medicinal Value*


DanBean is not only a highly nutritious natural protein source but also has excellent medicinal functions due to its abundance of bio-active compounds (seen in Figure 1). DanBean has multiple medicinal functions, such as antihypertensive and lipid-lowering functions, including certain curative effects on cardiovascular and cerebrovascular diseases [35]. DanBean is a rich source of the most prominent saturated fatty acid, α-linolenic acid, which lowers blood lipids and is also vital for anti-thrombosis and anticancer functions [44,45,46], Moreover, α-linolenic acid has an important impact on the intellectual development of infants and adolescents [47]. Polysaccharide CBP3 of DanBean has high antioxidant activity in vitro [43]. Liu et al. (2014) indicated that DanBean extracts have an anti-fatigue effect, making DanBean a very potent antioxidant and anti-aging food source [48]. DanBean has long been used as food, but more studies are warranted on its consumption and medicinal functions [42].

#### 2.2.3. Current Situation of DanBean Artificial Rearing

At present, some professional farmers are artificially cultivating two generations of DanBean every year. In spring, soybeans sown in the greenhouses as a food source to cultivate the first generation of DanBean, which is usually released to market during June and July. At the peak demand period, the DanBean price can reach up to 600 RMB/kg equivalent to 83 USD/kg, and the economic benefit is extremely considerable [49]. However, the corresponding rearing cost of DanBean is also high, thus there might be a trade-off between economic gains and high-tech establishment. Setting up a high-tech greenhouse during the preliminary rearing, costly DanBean eggs, and the strict technical requirements of rearing stages might seem challenging at the early stages of artificial rearing. The second generation of DanBean is reared with soybeans as a food source, where the soybeans are sowed as soon as the wheat is harvested. The DanBean is ready to be sent to the market in August and September. At this time, the market supply of DanBean is sufficient, and the price is much lower than that of the first generation available during June and July (100 RMB/kg equivalent to USD 14 or even lower) [49]. It is clear that the corresponding cost and risk of the second generation have decreased, but the profitability is much lower compared with the first generation of DanBean. However, in the Huang−Huai region, located in the eastern Guangdong Province of China, the low temperature in winter and spring is not suitable for soybean growth, thus limiting the rearing and product supply of DanBean during winter and spring. Therefore, DanBean can be mostly purchased from the market during June to September. In addition, some professional farmers have opened up rearing bases in southern provinces of China, and Hainan Province farmers have upscaled the rearing of DanBean. By utilizing the temperature and light resources, farmers plant soybeans to rear DanBean and can provide a small amount of fresh DanBean for the market even in the winter season. The availability of fresh DanBean during the winter season is very scarce, so the farmers can receive additional benefits if they can take the DanBean products to the market because of the high price, even at 83 USD/kg. So, the DanBean production in winter season boosts the economic benefits but the trade-offs are also very high [49].

The artificial rearing and production of DanBean includes egg acquirement, feed preparation, equipment design, environmental condition control, natural enemy control, and disease prevention, Among them, the feed cost accounts for 20–80% of the entire production cost. Therefore, feeding is of great significance for the production and utilization of artificial feed [50]. However, the exclusive feeding habit of DanBean (i.e., *C. bilineata tingtauica* larva) makes it difficult to wholly replace fresh soybean leaves with an artificial feed. Therefore the provision of suitable artificial feed is an indispensable prerequisite for the economical large-scale production of DanBean, needed to fulfill the exigencies of public consumption. Although information about feed formulation for the DanBean was mentioned in the ”*Manual of Artificial Insect Feed*” by Wang (1984) [51], the viability and applicability remains uncertain [36]. So far, there have been no reports of artificial rearing cases, so it is evident that the feed formulation practical manual for the artificial rearing of DanBean is more necessary than ever.

### 2.3. Silkworm Pupae

Silkworm pupae are known as original silkworm pupae or late silkworm pupae, and they are the pupae of *Bombyx mori* (Bombycidae: Bombyx) and *Antherea pernyi* (Saturniidae: Antheraea), belonging to Lepidopteran. These silkworms undergo complete metamorphosis and have the four distinct developmental stages of egg, larva, pupa, and adult (seen in Figure 4). Silkworm pupae are not only edible insects with rich nutritional functions but also a very valuable alternative source of animal protein and highly effective medicinal insects with multiple medicinal properties (seen in Figure 1).

#### 2.3.1. History and Status of Silkworm Pupae Consumption

In China, silkworms were domesticated early and used for silk reeling, and people have a long history of silkworm pupae consumption [52]. The edible and medicinal uses of silkworm pupae have been recorded in many ancient Chinese literary works, most notably mentioned by “*Qi Min Yao Shu*”, “*Historical Records*”, and “*Book of the Han Dynasty*”. According to the first two pieces of literature, silkworm pupa was listed as a delicious delicacy on the banquet table, while the latter mentioned the consumption of silkworm chrysalis protein where the pupae were consumed by adding salt after the cocoon underwent the reeling process. A Ming Dynasty medical scientist, Li Shizhen, systematically described the use of silkworm pupae as an edible insect food and medicine in his book “*Compendium of Materia Medica*” under the ‘Insects Section’. Li further mentioned that silkworm pupae were fried and eaten with spicy and salty flavors, as well as that these pupae were used as medicine in boosting immunity, regulating blood flow, reducing waist aches, strengthening vital organs like lungs and kidneys, and moistening the intestines. Chai et al. (2002) reported that long-term consumption of silkworm pupae can cure physical weakness, kidney diseases, and gastrointestinal malfunctions [53].

At present, fresh silkworm pupae are available on the market as raw food, snacks, and canned food. Hundreds of delicious delicacies are prepared from fresh silkworm pupae via frying, steaming, stewing, braising, roasting, and boiling. In Japan and South Korea, fresh silkworm pupae are savored with liquor. In the eastern Chinese provinces of Zhejiang, Shandong and Jiangsu, people have consumed fresh silkworm pupae since ancient times. For example, frying silkworm pupae with hot peppers and oil frying silkworm pupae with leek or tremella are common practices. In the northeastern Jilin Province, people used to prepare dozens of dishes with steamed and braised silkworm pupae. These natural and organic green foods cooked with fresh silkworm pupae are deeply loved by local consumers. These edible silkworm pupae come with diverse tastes (e.g., spicy, salty, sweet, sour and mellow), and they are also consumed as value-added products rich in aroma and texture. Eating silkworm pupae helps in the prevention of cardiovascular diseases, hepatitis, kidney deficiency, and emaciation for children [54]. Fresh silkworm pupae that are canned under different sterilization times and temperature conditions are also popular among consumers [55].

#### 2.3.2. Nutrition and Medicinal Value of Silkworm Pupae

Traditional Chinese medicine (i.e., TCM) believes that silkworm pupae have the functions of removing rheumatism, moisturizing the lungs and intestines, increasing the QI of the young, nourishing blood, strengthening the waist and kidneys, and treating infantile fever, wasting, dipsia, chronic hepatitis and a variety of cardiovascular diseases. Modern medicine (i.e., MM) believes that eating silkworm pupae can effectively increase human blood leukocytes with some pharmacological and biological active substances and their immunity, which is effective in treating hypercholesterolemia and also has the effect of improving liver function [56]. For example, Zhang et al. (2018) found that proteolytic peptide in silkworm pupae can significantly improve heart, kidney, and liver indexes in diabetic mice [6]. Zhu et al. (2008) pointed out that silkworm pupae extract has a certain dose-dependent effect on reducing blood sugar levels in diabetic rats and can effectively reduce blood sugar and blood lipid levels in diabetic rats. These results indicate that silkworm pupae have potential as hypoglycemic drugs or raw materials for health food [57]. Other studies have found that chrysalis peptide and the oil of silkworm pupae also have positive effects on lipid regulation, which can improve hyperlipidemia. Wang et al. (2009) extracted pupal oil from tussah silkworm pupae and carried out experiments in Wister mice, and the results showed that silkworm chrysalis oil and its unsaturated fatty acids could significantly reduce the contents of total cholesterol, triglyceride, and low-density lipoprotein in the serum of mice [58]. It can be seen that the silkworm pupa-related health products have huge development advantages and broad market prospects. It is suggested that combining modern pharmacology and cutting-edge technology could help to fully explore the role of chrysalis peptides and the oil of silkworm pupae in health care functions [59].

#### 2.3.3. Development Status of Silkworm-Rearing Technology and Industry

The silkworm oviposition control technology plays an important role in silkworm rearing. In the oviposition control process, to ensure that the application of silkworm oviposition technology is more reasonable, as well as to improve the comprehensive control effect of silkworm oviposition, the following aspects should be given more attention, as follows: (1) The silkworms need to finish mating one year before laying eggs, and special focus should be placed on monitoring the mating of female and male moths during the mating period. At the same time, the quantity and quality of moth mating should be controlled to ensure that the subsequent eggs can be effectively controlled. (2) After silkworms lay eggs, we should carry out the necessary management work to realize the comprehensive optimization of silkworm-rearing technology through egg management. During this management process, much more attention should be paid to the disinfection of silkworm rearing under a temperature of about 24 °C. (3) After silkworm oviposition, packing management is a vital procedure. At the same time, it is required to complete the analysis of the oviposition control technology [60].

In the past, the income of the sericulture industry (i.e., silkworm rearing) was at the forefront of the national agricultural and sideline production industry. However, in recent years, various tertiary industries have developed rapidly, and their benefits have exceeded those of the sericulture industry, which saw a rapid exit of the labor force from the sericulture industry and a flow of investments in efficient tertiary industries. Therefore, the enhanced artificial silkworm-rearing technology of the sericulture industry is much more difficult to promote due to the loss of a large number of experienced talents. This technical inadequacy gradually engulfed the artificial silkworm-rearing industry, robbing it off of the economic benefits. An unstable market and imbalance in the demand and supply chain is also a major risk in sericulture production. With the aim of developing the sericulture industry, promoting the latest artificial sericulture technology, improving the yield of mulberry plantation and silkworm cocoons, and strengthening the quality control of silkworm cocoons, it is vital to develop new technologies concerning the deep processing of by-products of silkworm cocoons (edible silkworm pupae) [45].

### 2.4. Wasps

Wasps, commonly known by their different names such as king bee, dragon bee, red-head bee, big-soil bee, Chinese tiger-head bee, and black-waist bee, belong to the Vespidae family of Hymenoptera, and they undergo complete metamorphosis and have the four distinct developmental stages of egg, larva, pupa, and adult (seen in Figure 5). There are about 15,000 species of wasps in the world, and more than 5000 species are known, of which 200 species of wasps have been recorded in China. Adult wasps are smooth and less hairy with longitudinally folded forewings at rest. Wasps are stinging insects with mostly blackish, yellowish, and brownish body colors with finely engraved reticulation, and they differ from bees through their distinct pointed lower abdomen and narrow waist, called a petiole, located between the abdomen and thorax. Wasps have long feet and are swift fliers with a strong labrum and mandibles, and they predate a variety of agricultural and forestry insect pests. Moreover, many wasps are popular edible insects with high nutritional value and therapeutic efficacy (seen in Figure 1), attracting a lot of new young connoisseurs in recent years. Therefore, the artificial rearing of wasps as an edible insect food not only increases income sources for farmers but also helps in the biological control of pests in the areas of agriculture and forestry [26].

#### 2.4.1. History and Current Situation of Wasp Consumption

In China, wasps are one of the most popular edible insects with a long-standing history of inclusion in diets. Chinese ancestors consumed wasp larvae and pupae nearly three thousand years ago, and the custom of consumption has been preserved in southwest regions since ancient times [61]. There is also the custom of eating wasp larvae and pupae outside of China; people from Mexico, Japan, and Thailand have included edible wasps in their diet and in paying homage to guests of honor [62,63]. In China’s Yunnan Province in particular, the custom of consuming wasp larvae and pupae as edible insect food is more common among different ethnic groups, and they are often cooked for guests as exotic delicacies [64]. The Yunnan Province was the core hub of such practices, which is well documented in the classical literature from the Tang Dynasty Era (618–907 C.E.) [65]. Nowadays, many ethnic groups in Yunnan Province still maintain plentiful methods of eating wasps, and the most common cooking method is directly frying wasp larvae and pupae with salt and pepper. In the Jinghong and Ruili regions of the Yunnan Province, the ethnic Dai people often boil wasp larvae and eat them with sour vinegar and spices [64]. According to the TCM mentioned in the book “*Shen Nong Ben Cao Jing*”, frequent consumption of edible wasps has been found to enhance skin tone, thereby resulting in a more youthful and attractive appearance [66,67,68].

#### 2.4.2. Nutritional and Medicinal Value of Wasps

The larvae, pupae, and adults of the wasp *Vespa basalis* are rich in amino acids and contain 18 types of amino acids [69]. The content of eight essential amino acids (EAAs) is 22.29–27.77%, which accounts for 41.21–61.44% of the total amino acids (TAAs), exceeding that recommended by the FAO/WHO Standard (EAA/TAA = 36%), and methionine (Met) is the first limiting amino acid in larvae, pupae, and adults of *V. basalis*, while the second limiting amino acid of pupae is lysine (Lys) [70]. In addition, the larvae, pupae and adults of *V. basalis* also have high contents of unsaturated fatty acids (54.76–68.88%), and the content of oleic acid (C18:1) is 35.80–55.90%, followed by palmitic acid (12.40–35.14%). The contents of two essential fatty acids (i.e., linolenic and linoleic acids), vital for the human body, were found to be 6.24–10.83% and 4.56–6.48%, respectively. Additionally, the contents of unsaturated fatty acids in larvae, pupae and adults of *V. basalis* ranged from 54.76% to 68.88%, so the wasps (*V. basalis*) are an ideal edible insect food source to supplement higher nutrition for human beings, featuring broad prospects and market development potential [70].

In addition, the larvae, pupae, adults, venom, and honeycomb of wasps can be used in TCM. The wasp venom has anticancer, antibacterial, anti-inflammatory, analgesic, antihypertensive, and radiation-inhibiting effects, as well as hemolytic and immunosuppressive effects [71,72]. According to the TCM book “*Shen Nong Ben Cao Jing*”, edible wasps have a sweet to neutral flavor, they are mainly used to treat headaches, eliminate the human body poison, and can also play a role in replenishing physical strength and restoring health. If wasps as an edible insect are consumed for a long time, it can delay aging [73]. In the TCM books of “*Compendium of Materia Medica*” and “*Color Medicine Map Compendium of Materia Medica*”, it was mentioned that edible wasps found in cold climates have a sweet taste, are slightly toxic, and have many medicinal properties like preventing abdominal distension and retching, detoxification elements, as well as the potential to remove freckles and facial sores [74]. The Jingpo people, an ethnic group of the Yunnan Province, commonly use fresh wasps to prepare wasp wine, which has the effect of removing dehumidification. It is often used in the treatment of acute rheumatism, rheumatoid arthritis, and joint pain [64]. Wasps incorporate propolis, beeswax, and resin into their nesting environments, and edible wasp larvae are also rich sources of trace elements like copper, iron, and manganese, as well as vitamins like riboflavin, biotin, and folic acid. Simultaneously, edible wasps include several pharmacological properties, such as anticancer, hypoglycemic prevention, hypolipidemic prevention, anti-inflammatory, antibacterial, and antiviral elements. At the same time, they also have therapeutic effects on cardiovascular diseases [75]. In addition, the coating agent made of wasps can prevent and protect against cerebral ischemic injury [76], relieve pain, including anti-coagulation and anti-thrombosis properties [77], and resist hypoxia and myocardial ischemia [78]. So, the medicinal value of wasps is extremely high, and they have many excellent uses in TCM.

#### 2.4.3. Current Situation of Wasp Artificial Rearing

In recent years, using wasps as edible insect food has gained momentum, which provides a bright prospect for the rapid market development of edible wasps and will accelerate the current trend of artificial rearing of wasps for the scaled-up production of edible wasps as a dietary food. In the beginning, the artificial rearing practice is followed by using dead insects or pork as bait to trap wasps from June to July every year. After trapping the wasps, a white feather is tied to the wasp’s waist with a thin thread or plant fiber, and it is tracked until the honeycomb is found. The honeycomb is retrieved at night and placed in a controlled environment for wasp rearing. Due to the lack of understanding of the biological habits of wasps in the early rearing stage, the limitation of rearing density and location often leads to the failure of artificial wasp rearing.

In the 1970s, wasps were used mainly for controlling natural enemies of agricultural pests. In 1974, *Polistes hebraeus* and *Vespula germanica* were used to control cotton insect pests in Shangqiu County of the Henan Province. Later, people began to study the artificial rearing of wasps, and great progress was made in the management and nesting after overwintering [79,80]. Liu (1997) summarized the artificial rearing technology of medicinal wasps [81]. In the early exploration period, the artificial rearing technology was imperfect and a systemic rearing system has still not been formed to date. During the 1990s, the large-scale rearing of wasps was primarily explored and achieved success [82]. Subsequently, Zheng and Tan (2008) studied the artificial rearing technology of *V. velutina* and established a relatively perfect rearing technology system which laid the foundation for the large-scale rearing of wasps [83]. Later, the artificial large-scale rearing technology of wasps was further explored, and the procedures of wasp mating, overwintering, nesting, and primary swarm management were explained, which promoted the artificial rearing of wasps [84,85,86]. At present, artificial wasp rearing exists mainly in south China, especially in the Yunnan Province, followed by the Sichuan, Guizhou, Guangxi, Guangdong and Hunan provinces. The main species of rearing wasps is *V. velutina*, accounting for more than 90% of the total rearing, followed by *V. mandarinia*. In addition, some farmers have explored the cultivation of wasp species such as *V. basalis* and *V. affinis*, but the success remains unfolded [87].

### 2.5. Locust

Locusts, commonly known as grasshoppers (Orthoptera: Acrididae), have more than 10,000 species in the world, and they distribute in tropical regions, temperate grasslands, and desert areas. Locusts undergo incomplete metamorphoses and have the three distinct developmental stages of egg, nymph, and adult (seen in Figure 6). There are more than 1000 species of locusts in China, and the most predominant ones are *Locusta migratoria manilensis*, *L. migratoria*, and *L. migratoria tibetensis*. Among them, *L. migratoria manilensis* has the widest distribution range and is considered the most destructive one; it is responsible for the locust plague in China, mainly endangering grasslands, and is a major agricultural pest. In the “*List of Class I Crop Diseases and Pests* (2023)” released by the Ministry of Agriculture and Rural Affairs, locusts rank second as the most destructive insect pests in China. Locusts are usually green, gray, brown, or dark brown in appearance, and their strong and well-developed hind legs enable them to gain momentum during flight. Locusts are herbivorous insects with gregarious feeding habits, feeding on leaves and the tender tissues of plants, and they are generally solitary or migratory and scatter in diverse habitats and migrate in flocks for crop damage. The vast majority of locusts survive in soil with a life span of 75 days [88].

#### 2.5.1. History and Current Situation of Locust Consumption

Locust swarms are known for the destruction they can cause to vegetation and agriculture, but here we mainly discuss their beneficial aspect as edible insects. Of the 21 known species of locusts, around 10 species have been traditionally consumed by humans or fed to animals in 65 countries for thousands of years [89]. Locust eggs, nymphs and adults are all edible, and they have a significant place in the hierarchy of edible insects [90]. Radiocarbon dated to 2600 BC suggests us people in Libya began to consume roasted locusts as food [91]. Red locusts are consumed in Mexico, Kuwait, and Saudi Arabia [92]. Migratory and Bombay locusts are a delicacy in many Asian countries (China, Thailand, Japan, Philippines, Vietnam, India, Laos, Malaysia and Myanmar), South America (Mexico), and Europe (Ukraine and Belgium) [92,93,94,95]. Additionally, the Australian plague locusts are enjoyed in Australia, the Italian and Siberian locusts are enjoyed in China, and the South American locusts are enjoyed in Brazil and Colombia [92]. Food patterns and the types of locusts eaten vary from country to country. Among them, the Japanese people like to consume locusts through cooking and frying. Kim et al. (2020) reported that *Oxya chinensis*, a species of locust seen in Chinese rice fields, has also emerged as a favorable edible insect species with exceptional nutritional qualities and as a potential nutraceutical for anticancer immunity improvement [96]. Mexico is known as the living reservoir of insects, where more than 20 species of locusts are found, and the most popular locusts belong to *Sphenarium* sp., which are the most widely distributed and have antioxidant properties. In Thailand, “fried flying shrimp” has become the most favored popular health food for the local people, and, as early as 1983, the government stipulated that locusts that harm crops could only be caught artificially and that it was strictly forbidden to spray pesticides because of the consumption habit. American consumers have initiated modern eating methods by savoring fried locusts, snacks, and other pleasant delicacies [97].

China also has a long history of consuming locusts. During the locust plague in China, people consumed locusts to combat famine, but now people eat locusts as a highly nutritious protein source [98]. Now, with the improvement of people’s living standards, locusts have long since become a delicacy in high-end restaurants [99]. Artificial rearing of locusts in a plastic tunnel can produce dozens of tons of locusts a year through small farming enterprises [100]. Farmers harvest locusts, package them and send them to restaurants in the city [26]. In Tianjin City, there has been a tradition of eating locusts as a snack since ancient times. Ethnic minorities in mountainous areas of the Guangxi Province celebrate the annual “Insect-Eating Festival” on the second day of June. Every family prepares a variety mouth-watering insect meals, and locusts are naturally among them, while the most consumed delicacies include “Fried locust” and “Pickled ant” [98].

#### 2.5.2. Nutritional and Medicinal Value of Locusts

Locusts are rich protein sources, and they have higher protein content than meat [101]. Chitin can be obtained from locusts, which are used for medicinal and biomedical purposes [102]. Locusts not only have high nutritional value but also have high medicinal value. The use of locusts in TCM has been listed under the name “*Zha Pan*” in the book “*Supplement to Medica*”, where the medicinal properties of consuming locusts have been described in detail. Locusts as an edible insect food alone or in combination can treat a variety of medical abnormalities, such as tetanus, acute and chronic convulsions, measles, fever, asthma, bronchitis, and cardiovascular and cerebrovascular diseases. It also has the effects of lowering cholesterol, reducing blood pressure, and weight loss. Long-term consumption of locusts not only has the health function of relieving coughs and asthma, enhancing strength, and invigorating the spleen, but it also can prevent the occurrence of cardiovascular and cerebrovascular diseases [71].

The locust *Schistocerca gregaria* contains important sterols (e.g., β-sitosterol, rapeseed sterol and stigsterol), which help to combat hypertension and cardiovascular diseases [103,104]. Hydrolyzed *S. gregaria* protein can inhibit angiotensin-converting enzyme (ACE), which converts inactivated angiotensin-I into octapeptide angiotensin-II and inactivates the vasodilator bradykinin, resulting in increased human blood pressure [105]. Therefore, locusts can be used as antihypertensive components in nutritional and health products or alternatives to synthetic ACE inhibitors (e.g., captopril), which can cause serious side effects such as cough and angioedema [105]. The hydrolyzed products of cooked and baked *S. gregaria* can proliferate human skin fibroblasts, which are essential for synthesizing components of the skin’s extracellular matrix [106]. External signs of aging arise with advancing age or other factors that block fibroblast proliferation; therefore, including edible locusts in daily diets can reduce the signs and symptoms of aging.

#### 2.5.3. Locust-Rearing Technique

Locust swarming is a serious threat, and scientists are still searching for scientific and practical ways to mitigate such pernicious insect outbreaks in agriculture. On the one hand, locust-rearing can pose an everlasting threat to agriculture and the whole ecosystem due to the increased density of locusts; however, on the other hand, there is also a pathway for farmers to gain additional income from artificial locust-rearing. Therefore, it is necessary to take utmost precautionary measures so that the ecosystem services are not threatened by locust attacks and so farmers can supply the edible locusts to the market and receive additional income. In the process of locust-rearing, a high density of reared locusts should be prevented, and the rearing tunnel should neither be too dry nor too wet [100]. For large-scale rearing, it is necessary to set a 0.5 m wide plant infection isolation zone around the farm where it will be convenient to observe whether the greenhouse is damaged or not. Pesticides and all the necessary equipment should also be well-equipped prior to locust feeding. If the locusts are to be no longer cultivated, they need to be thoroughly destroyed. Before mid-September (before the locusts have mated), all locusts will be sold or killed with pesticides, eliminating their potential hazards and escorting the healthy development of the locust farming industry [100].

It is possible to use the initial provenance of locusts in any state, but this requires ongoing selection, domestication, and eradication in terms of provenance technology. Based on successful locust-rearing local scenarios, and sticking to the procedures mentioned by Liu (2005) [107], here we have mentioned the procedures for the rearing of *Oxya chinensis*, one of the dominating locust species in China predominantly seen on the rice fields of the Yangtze River Delta. The details of the locust-rearing technique are given below: (1) Well embanked and untilled wet rice field with weeds left open and unturned soil are the ideal habitats for locust laying eggs. The temperature should not exceed 30 °C, and relative humidity should be maintained at 60%. (2) Then, the rice field ridges are carefully dug to preserve the egg blocks. Locusts deposit their egg pods in the soil at a significant depth (about 10 cm). The bins for laying eggs must be sufficiently deep. The eggs are hatched after 10–15 days. (3) The feeding of locusts is a must for their proper growth and development, so the feed cultivation technology relies on the host plant. For the artificial rearing of *O. chinensis*, we need to select the rice field. *O. chinensis* mainly feeds on rice, but it can also feed on the stems and leaves or grains of gramineous or sedge plants such as corn and beans. They can also be fed on the grasses of *Cynodon dactylon* and *Sorghum halepense*. Generally, the above-mentioned plants, with lengths of about 30 cm, are used, and they are scattered in the feeding box and carefully fed every morning. Food consumption by locusts is typically minimal in high air humidity conditions. In addition, the feeding boxes should be kept clean and the locust feces and residues clearly removed. Thus fed locusts, when they are fully developed to adults, are sent to the market for selling after careful harvesting.

We summarize the locust-rearing procedures into four key points: (1) *Construction and completion of rearing shelter.* Before the construction of the shed, the ground ants and mole crickets should be eliminated cleanly. Monocot plants on the ground are used to rear young locusts [108]. (2) *Egg hatching.* The temperature and humidity must be controlled when the incubation temperature is 28~32 °C, and the humidity requirements of locust eggs are not very strict [109]. (3) *Management before and after locust spawning.* The spawning site must be kept clean to facilitate the spawning of female locusts [108]. At this time, locusts undergo the key period of physical consumption, so we must ensure sufficient foraging grass, and the humidity can be controlled at 15–17%. In addition, the light requirements should not be ignored and should be kept at 16 h day time [109]. (4) *Feeding management from the third instar to the adult stage.* After the third instar, the flight speed and food intake of the nymphs gradually increases. This is a critical period for the growth of locusts, so the adequacy of food directly affects the normal growth of locusts [109].

#### 2.5.4. Development Status of the Locust-Rearing Industry

The common species of edible locusts are *L. migratoria manilensis*, *O. chinensis*, *Acrida cinerea*, and *Chondracris rosea* [107,110]. At present, the development of large-scale rearing of locusts in China is uneven, though it has been a successful enterprise in rural counties. There are about 1000 locust farmers of different sizes in China. There are about 20 = 30 large-scale and influential companies, with annual locust production ranging from few kilograms to dozens of tons and locust farming areas ranging from dozens of square meters to tens of thousands of square meters [111]. Large scale locust-rearing bases with high influence are mainly distributed in Hebei, Shandong, Liaoning, Hubei, Tianjin, Beijing, Guangxi, and other provinces and cities. Among them, the Hebei and Shandong provinces have the largest number of locust-rearing bases. China’s annual output of locusts is hundreds of tons, which mainly supplies hotels and locust processing plants in large and medium-sized cities [111]. The edible locust *O. chinensis* has seen a recent up-rise in exports from China to Japan, Australia, South Korea, Germany, and the United States [90,112]. It is important to note that, if the artificial rearing technology of rice locusts *O. chinensis* is technically enhanced for large-scale commercial production, the farmers will tremendously benefit.

### 2.6. Insect Products Used as Edible Insects

Common insect products that can be gathered from edible insects include royal jelly, bee venom, and silkworm silk, all of which are high in nutrients and well known for their pharmacological effects. Among them, royal jelly is a creamy substance secreted from the mandible and hypopharyngeal glands of worker bees; it is the food of all bee larvae in the first 1–3 days after hatching and is composed of bee bread (fermented pollen + honey), while the queen bees can feed on the royal jelly during their entire life span [113]. Royal jelly is highly nutritious and is also recognized as a beneficial nutrient for human health, so it is loved by the majority of consumers, praised as “super-food”, and commonly sold in functional health products and cosmetics. Royal jelly and its derivatives have a very wide range of medicinal values where they are widely utilized for the treatment and prevention of human health and disease. According to Liu et al. (2021), royal jelly is currently utilized for the control of “3H”, diabetes, cancer, neuromodulation, geriatric diseases, and diseases associated with the reproductive and endocrine systems [114]. It also has a protective effect on the cardiovascular, nervous system, digestive system, respiratory system, and excretory system [114].

In addition, there are other insect products, such as bee venom and silkworm silk. Bee venom is a colorless and transparent liquid with an aromatic odor secreted by the venom glands and accessory glands of worker bees, and this vital bee product has a long history of applications in China [115]. Bee venom has been used in the medicine, health care, and cosmetics industries due to its anti-bacterial, anti-inflammatory, anti-tumor, hypoglycemic, and hypolipidemic qualities, and for its protection of the liver and nervous system [116]. Silkworm silk is a pure protein source due to the high contents of protein, amino acids (especially alanine), and cellulose, and it can be processed and dissolved into colorless tasteless “edible silk”. The utilization of edible silk offers a plethora of health benefits, such as the enhancement of liver function, the reduction in cholesterol levels in the bloodstream, the elimination of toxins from the human body, the prevention of constipation, and the facilitation of a decrease in alcohol content in the blood [117]. As a result, silkworm silk can be effectively employed as a novel type of health food (e.g., delectable candy, noodles). Moreover, their suitability to the production of food items specifically tailored to the dietary requirements of elderly individuals is highly noteworthy [117].

## 3. The Main Nutrients and Active Ingredients of Edible Insects

### 3.1. Proteins

#### 3.1.1. Type and Content

Edible insects are rich sources of proteins and also act as micro-nutrient reservoirs. Out of the nearly 100 insect species that have been analyzed for their nutrient content, the crude protein content generally ranges from 20 to 70% regardless of the insect form [30]. The protein content of edible insects is significantly higher than that of plant foods in general, and it is even higher than that of industrial livestock and poultry products (e.g., fish, eggs and dairy products). Wang et al. (2011) found that the protein content of the grass katydid/cone-headed grasshopper, *Euconocephalus nasutus*, the black-shinned hook-topped katydid *Ruspolia lineosa*, and the skinny cone-head katydid *Pyrgocorypha gracilis* were 69.38%, 70.38% and 61.79%, respectively [118]. The protein content of adult kiwi wasps *V. basalis* was 71.07%, while the protein content of golden-bordered ground turtle *Opisthoplata orientalis* was as high as 80.00% [70,119]. The protein content of domestic crickets was relatively high, ranging from 41.8 to 72.33 g/100 g [120]. In contrast, the protein content of plant-based foods (e.g., soybeans and peanuts) was 38.0% and 24.7%, respectively [121].

Some studies have proven that the protein content of edible insects in different forms is different, i.e., the highest in adults, followed by pupae, and the lowest in larvae. The protein content of the adult, pupa, and larva of *V. basalis* was found to be 71.07%, 58.59% and 50.83%, respectively [70]. In addition, a similar pattern was found in edible insects such as barley bug *Zophobas atratus*, nasal euglena katydid *Euconocephalus nasutus* and thin conehead katydid *Pyrocorypha gracilis*. Additionally, the protein content of different species of edible insects varied as follows: Orthoptera > Homoptera > Dragonflies > Diptera > Hymenoptera > Hemiptera > Lepidoptera > Coleoptera [62].

It is worth noting that the nitrogen−protein conversion coefficient of 6.25, often used to estimate insect protein, is faulty because it overestimates the protein content of flying locusts and other insects by about 17% due to the chitin content [122]. Boulos et al. (2020) suggested a conversion coefficient for “nitrogen−protein” of 5.33 to be more accurate in estimating the protein content of locusts and other edible insects consumed as food [122]. On the other hand, not only will the morphology and structure of insects change greatly at different development stages, but it will also be affected by physiological and metabolic changes within the body, which will lead to differences in the content of major nutrients such as proteins and active substances [121].

#### 3.1.2. Nutritional Value

Not only is edible insect protein now the traditional dietary habit, the search has also led to the normalization of the consumption of edible insects, bringing enormous nutritional and medicinal benefits. Eating edible insect products is conducive to improving the human immune system and triggers anti-aging elements. It has great benefits for patients with “3H” (i.e., hypertension, hyperlipoidemia and hyperglycaemia), and even contributes to recovery from severe diseases like hepatitis B [123]. After the enzymatic hydrolysation of yellow mealworm *T. molitor*, mixing with the amino acid solution, and after a series of production steps, functional nutrient drinks can be manufactured. The incorporation of yellow mealworms *T. molitor* pupae significantly reduces the specific volume of bread and significantly increases the content of crude protein, thus improving the nutritional value of bread [124]. The larvae of the domestic silkworm *B. mori* have anti-gastric and anticancer effects [124]; while the protein-rich substances isolated from the larvae of the domestic fly *Musca domestica* can inhibit H_9_N_2_ infection and kill the polygosome virus in alfalfa [125].

#### 3.1.3. Special Protein with Nutrition Function

Medical research has shown that the pupal protein in the edible silkworm *B. mori* can reduce the risk of many diseases, including cardiovascular diseases (hypercatatonia and hyperlipidemia) and cancer (liver and stomach cancer). In addition, enzymatic hydrolysates of silkworm (*B. mori*) pupae have been reported to be biologically active and aid in antioxidant activity and immune regulation activity, and they can improve hypercholesterolemia and anti-tumor activity [126]. Pan et al. (2022) further reported that, after the enzymatic digestion, the reduced silkworm pupal protein also can isolate umami peptides from the proteolysis, which can be used as an important source of dietary protein, thereby providing additional benefits for human health. In addition, many insects can also produce antibacterial peptides (antibacterial protein), interferon, and other special proteins with obvious tumor suppression effects [126].

### 3.2. Lipids

#### 3.2.1. Type and Content

Edible insects are rich in fat, with the fat content generally ranging from 10% to 50%; among them; the fat content of pupae and larvae is higher, and that of adults is lower [127]. Related studies have also shown that the fat content of edible insects varies among different insect orders, mainly as follows: Coleoptera > Lepidoptera > Hemiptera > Hymenoptera > Dragonflies > Diptera > Homoptera > Orthoptera [62]. Unlike the fat composition of animals in general, edible insects are rich in unsaturated fatty acids and palmitic acid, which are comparable to fish [128]. The two polyunsaturated fatty acids (linoleic acid and linolenic acid) are necessary for human growth and development. Edible insects are rich in a high proportion of polyunsaturated fatty acids, and Lepidopterans have a high linolenic acid content, while linoleic acid content is higher in Coleoptera [127]. Zhang et al. (2018) pointed out that the proportion of unsaturated fatty acids in the pupa of non-mulberry silkworm *Antherea pernyi* is more than 80%, the proportion of unsaturated fatty acids in ants *Pheidole megacephala* is about 80%, and the proportion of unsaturated fatty acids in termites *Macrotermes subhyalinu* is about 74% [6]. Xie et al. (2022) indicated that the content of unsaturated fatty acids in the yellow mealworm *T. molitor* is also notably high [129]. Siddiqui et al. (2024) indicated that the domestic cricket *A. domesticus* contained an appropriate amount of fat ranging from 4.30% to 33.4% of body biomass, while *A. domesticus* is also rich in fatty acids. It can be seen that the consumption of insects in a daily diet can supplement humans with rich unsaturated fatty acids, especially polyunsaturated fatty acids [130].

#### 3.2.2. Nutritional Value

The most prevalent sterol in insects is cholesterol, and edible insects have lower levels of cholesterol than conventional protein sources like eggs. The cholesterol content of fresh raw eggs was 372 mg/100 g, three times more than that of domestic crickets (105 mg/100 g), Bombay locusts (66 mg/100 g), and scarab beetles (56 mg/100 g) [131]. Therefore, edible insects are a great potential source for manufacturing foods with high nutritional value and low cholesterol content that can mitigate the “3H”. In addition, unsaturated fatty acids abundant in edible insects have important physiological functions, such as lowering blood lipids, lowering blood cholesterol levels, and inhibiting the generation of peroxidized lipids in the body.

#### 3.2.3. Nutritional Functions of Special Lipid

Phospholipids are important components of the protoplasm of animal and plant tissue cells. Tian (2015) mentioned that the phospholipids in insect lipids mainly include phosphatidylcholine, phosphatidylethanolamine, and myoinositol. They are involved in fat metabolism in the human body and has effects on brain health, lowering blood fat, removing cholesterol, treating fatty liver and cirrhosis, and having anti-aging effects. Lecithin in Gossypium, Lepidoptera, Hymenoptera, and vertebrates is dominated by phosphatidylcholine, but in Diptera it is dominated by phosphatidylethanolamine [132].

### 3.3. Amino Acids

#### 3.3.1. Type and Content

Edible insects are rich in nine essential amino acids and 17 other amino acids, with the highest content of glutamate (Glu) being higher than those of all animals. The amino acids present in edible insects are not only abundant but also well balanced, making them a high-quality source of amino acids [15]. The proportion of amino acids in most edible insects is close to or exceeds the amino acid requirement proposed by the World Health Organization (WHO) and the Food and Agriculture Organization (FAO) [128], which is conducive to human and animal absorption [6]. The main limiting amino acids in edible insect proteins are tryptophan (Trp) and threonine (Thr), while the main limiting amino acids in fish and livestock meat proteins are methionine (Met) and valine (Val), and the main limiting amino acid in cereal protein is lysine (Lys) [133]. Therefore, edible insects can well complement plant food and animal food in supplying essential amino acids; for example, tussah pupae are rich in Lys (mass ratio of 0.0654 g/g), which can help to compensate for the deficiency of Lys in grain protein [134].

#### 3.3.2. Nutritional Value

The amino acid digestion utilization of edible insects can reach 98.93%, and the lowest value is over 70%. The digestibility of meat and fish, which is close to or above the amino acid digestibility, is significantly higher than the digestive utilization rate of plant protein [135]. The amino acids contained in edible insects, in addition to protein synthesis, are involved in several specific metabolic reactions in the human body, which are mainly manifested in some important properties of the regulation of physiological functions, such as the dilation of blood vessels, lowering blood pressure, inhibiting cancer cell proliferation, and preventing memory disorders [17]. The human body uses amino acids for the synthesis of proteins as well as several other biological processes and metabolic reactions. For example, His dilates blood vessels and lowers blood pressure [132]. Limited to cereal proteins, Lys is an essential amino acid that helps preserve the animal’s acid-base balance, facilitates the absorption and use of other amino acids, and safeguards the nervous system [136]. Therefore, consuming edible insects can provide richer amino acid nutrition for human health and has a very high nutritional value.

### 3.4. Special Carbohydrates

#### 3.4.1. Alginate

Abundant sugar is present in edible insects; in addition to glucose, fructose and glycogen, insect hemolymph also contains a significant amount of alginate, which has enormous health benefits. According to Jiang et al. (2016), the growth factor of bifidobacteria, a helpful gut bacterium, alginosis can enhance the microecological environment of the intestine, fortify the gastrointestinal tract’s ability to digest and absorb nutrients, efficiently remove toxins from the body, and boost the immune system’s resistance to pathogenic organisms. Furthermore, alginose has a potent anti-radiation effect [46].

#### 3.4.2. Chitin

The second most common polysaccharide in nature is chitin, and it is a long-chain polymer of N-acetylglucosamine, an amide derivative of glucose. Insect exoskeletons are primarily composed of chitin, which serves as their fundamental building block. Edible insects are rich sources of chitin. The deacetylation of chitin, which is derived from insects and crustaceans, yields chitosan as an amino polysaccharide. The chitosan content in the body of insects is between 5 and 15%, while that of the adult of Yunnan pine caterpillar *Dendrolimus houi* is up to 17.83% [30]. The yellow mealworm *T. molitor* is also rich in chitosan [129]. Many health benefits can be obtained from chitosan, a dietary fiber that is difficult for humans to digest and absorb after consumption. These benefits include promoting peristalsis in the digestive tract, adsorbing toxic substances, lowering blood pressure, intra-intestinal pressure and abdominal pressure, preventing intestinal cancer, and lowering blood cholesterol [17].

#### 3.4.3. Cordyceps Polysaccharide

*Cordyceps sinensis*, a parasitic fungus to the moth larvae (Lepidoptera) of *Hepialus armoricanus*, is known for its polysaccharides which are the focus of research and quality assurance in *C. sinensis* health products [137,138]. These polysaccharides can be categorized into two groups based on their location within the fungal cells: intracellular polysaccharides (IPSs) and extracellular polysaccharides (EPSs). IPSs are obtained from the fruiting body (or worm) and mycelium of *C. sinensis*, while EPSs are extracted through a centrifugation process [139,140]. Cordyceps polysaccharide is a kind of complex organic macromolecule, and its combination with proteins will form glycopeptides. *C. sinensis* lives parasitically on bat moth larvae and *Cordyceps chrysosporium* lives parasitically on the cocoons and pupae of Lepidopteran larvae, while the Cordyceps polysaccharides produced by both of them have a wide range of pharmacological effects, such as anti-tumor, anti-osteoporosis, hepatoprotective, renoprotective, and immune-modulating elements [141,142]. Some studies have shown that Cordyceps polysaccharide and *C. chrysanthemi* polysaccharides have medicinal properties, such as antioxidant and promoting immunomodulation, as well as anti-tumor activities and kidney and liver protection elements [143]. In addition, Cordyceps polysaccharide involves biological activities, such as lowering blood glucose and blood lipids, anti-radiation, and delaying aging aspects [144,145].

### 3.5. Other Nutrients and Bio-Active Compounds

#### 3.5.1. Minerals

The minerals include essential macro and trace elements as well as non-essential heavy metals. Macro and trace elements are required for normal growth and physiological functions of the body, and they should be supplied in sufficient quantities through food. The alternative protein sources, edible insects, act as mineral repositories, storing minerals that are more abundant in zinc and iron than traditional livestock meat products (e.g., beef, pork, and chicken). The Chinese rice locust *O. chinensis* contains nine different types of elements, mainly including Na, Mg, Fe, Zn, and Se [146]. Ants have a higher content of Zn than that seen in soybeans, and twice as much as pig liver, and they are also rich in 28 different types of essential minerals, mainly including Mn, Zn, Se, Mg, Ca, P, and Fe [147]. Live golden cicada nymphs are comprised of 58.58% protein, 10.23% fat, 0.58% total phosphorus, 16.0 mg/kg calcium, and 82.2 mg/kg Zinc [33]. It is worth noting that Fe deficiency is dangerous for anaemic individuals, and Zn deficiency is detrimental to the body’s anti-ageing process. Therefore, the consumption of edible insects can replenish essential minerals, such as Fe and Zn, which are useful in treating anaemia and slowing down the aging process.

#### 3.5.2. Vitamin and Carotenoids

Edible insects are rich in vitamins and carotenoids., and vitamins A, B_1_, B_2_, B_6_, D, E, C, and K are relatively high among them. Qiao et al. (1992) demonstrated that the rice locust *O. chinensis* contained vitamins A, B1, B2 and E, as well as carotenoids [146]. Xu and Xia (2004) reported that the soil edge termite *Macrotermes annandalie* constituted 25 IU/g and 85.4 IU/g of vitamin A and D, respectively, the vitamin B_2_ substance within the pupae of *T. molitor* and *O. chinensis* was 5.8 mg/kg and 16.2 mg/kg, respectively, and the vitamin B_2_ substance within the crisply squeezed pupae powder of residential silkworms *B. mori* was as high as 63.92 mg/kg [17]. The TCM “silkworm sand”, which is the dried feces of the domestic silkworm larvae of *B. mori*, contains vitamin E and β-carotene and has the benefits of removing dampness, metabolic stability, and resolving turbidity, as well as invigorating blood circulation and circulation of menstruation [148].

#### 3.5.3. Specific Active Substances

In addition to the above-mentioned proteins, lipids, amino acids, and special carbohydrates, edible insects also contain many specific active substances (seen in Table 1). Liu et al. (2004) reported that Hymenopteran insects like termites and ants contained some special proteins and interferon, which had tumor-inhibiting effects [149]. Wang et al. (2004) found that many insects (e.g., *Hyalophora cecropia*, *Manduca sexta*, *Sarcophaga percgrina*, *Phormia terranovae*, *Drosophila melanogaster* and *Zophoba atratus*) can produce antimicrobial peptides, which have significant tumor-inhibiting effects [150]. Xu and Xia (2004) found that the chitin in *D. houi* had the health function of dietary fiber, and it also had hemostatic and anti-thrombotic functions [17]. Feng (2005) reported that the content of chitin in the nymphs and adults of *D. houi* was 7.47% and 17.83%, respectively, and the lecithin content in the eggs of *Ericerus pela* was 9.22%, which had the functions of enhancing the immunity of the human body and the prevention of atherosclerosis, hypertension, heart disease, and fatty liver [151]. Liu et al. (2008) also showed that the flavonoid extract of *O. chinensis* had hypolipidemic, anti-fatigue, and antioxidant effects [152]. Tian (2015) indicated that phospholipids had the functions of brain health, lowering blood lipids, removing cholesterol, treating fatty liver and liver cirrhosis, and preventing aging [132]. Jiang et al. (2016) proved that alginose in insects had a strong anti-radiation effect that would help to improve the intestinal microecological environment, strengthen the gastrointestinal tract digestion and absorption function, effectively eliminate toxins in the body, and enhance the human body’s immune resistance to disease [46]. Zhong et al. (2017) found that Cordyceps polysaccharide had biological activities such as lowering blood glucose and blood lipids, anti-discharge elements, and delaying aging [144]. Chen (2020) also found that Cordyceps polysaccharide exhibited significant antioxidant activity and immunomodulatory effects [143]. Hall et al. (2020) found that anti-inflammatory peptides could be extracted from protein hydrolysate as angiotensin-converting enzyme inhibitors from crickets (*Gryllodes sigillatus*) [153]. Pan et al. (2022) found that the consumption of pupal protein of the silkworm *B. mori* was associated with a reduction in cardiovascular disease, cancer, and other disease risks by collating and summarizing medical studies, and they concluded that enzymatically produced hydrolysates of silkworm pupae have biological activities, e.g., antioxidant, anti-tumor, and immunomodulatory activities [126].

## 4. Resource Utilization Advantage of Edible Insects

### 4.1. Characteristics of Edible Insect Production

Edible insects possess abundant resources that offer a reliable foundation for rational utilization by humans, offering an array of benefits ranging from alternate protein sources to pharmacological attributes [22]. The cultivation of edible insects showcases both natural and artificial attributes. The economically valuable edible insects, abundant in nutrients, minerals, trace elements, and secondary metabolites, offer great potential for development and utilization. Edible insects are considered a wholesome source of nutrition, being completely natural and free from any synthetic chemicals or additives. Compared to the tedious manual production process, edible insect products are safer, and the production process is simple, which can guarantee food safety [154]. Compared with the directly fried edible insects (locusts, golden cicada and coconut worms), artificially extracted and processed edible insect food (insect compound amino acid beverages and high-protein nutrition snacks) is much more likely to be accepted by consumers [155]. The characteristics of the edible insect production process include a fast reproduction speed and low feeding cost. Nevertheless, a crucial question arises as to whether edible insect products are naturally sourced or artificially created. Firstly, this is determined by the biological characteristics of the growth and development of edible insects. Secondly, edible insects are widely favored by consumers because of their unique nutritional value and healthcare functions. There are many species of edible insects, and each species has different functional factors. For example, the functional factors of ants are Zn and grass formic acid, and the functional factors of *C. sinensis* are mannitol, cordyceps polysaccharide, SOD, and selenium. Functional factors of gallnut include tannic acid and gallic acid. The functional factors of silkworm excrement are chlorophyll [17]. Several studies have provided evidence that the functional components found in edible insects are entirely natural and do not accumulate any harmful substances within the human body [17].

### 4.2. Utilization Methods of Edible Insects

#### 4.2.1. Direct Consumption

Consuming edible insects in their natural form involves treating and cooking them without the use of industrial processes in order to preserve their original characteristics. The conventional approach involves transforming edible insects into various culinary preparations, such as fried locusts [97], stir-fried silkworm pupae [53,54,55], cicadas [31,33], ants [12,13], and so on. The increasing curiosity of individuals has led to the creation of various delicious dishes using edible insects, indicating promising market opportunities. Nevertheless, the unappealing appearance of most edible insects and their below-par living conditions have hindered widespread acceptance of insect-based cuisine.

#### 4.2.2. Consumption of Processed Edible Insects

Due to the advancements in modern science and technology, edible insects have been transformed into processed insect foods such as biscuits, bread, wine, soy sauce, and amino acid-based oral liquids. Currently, the spotlight is on insect protein food in the diversified expansion of edible insect cuisine [156]. The pure protein extracted from edible insects can be utilized as essential nutrients or fortified food supplements and can be added to different food items like bread, biscuits, noodles, pastries, and tonic beverages [157]. For instance, the utilization of silkworm protein involves its conversion into protein powder, which can be conveniently incorporated into various food products as a valuable source of protein nutrition. This protein powder is particularly beneficial for infants and the elderly, as it serves as a nourishing dietary option [158]. In China, the utilization of silkworm protein powder has been advanced to manufacture high-protein biscuits, like ‘henglibao ant powder’ [159]. This innovation has the potential to alleviate individuals’ sensory bias and apprehension towards consuming insects by creating protein-rich food products derived from edible insects [160]. Consequently, this development holds significant advantages for the widespread acceptance and promotion of edible insect-based food [15].

#### 4.2.3. Consumption of Nutritional and Health-Care Products Made from Edible Insects

Traditional Chinese medicine theory has played a significant role in shaping China’s increased interest in the pharmacological effects of edible insects. This has led to the establishment of a considerable industry dedicated to extracting nutrients from insects, particularly for manufacturing nutrition and health products [159]. In provinces such as Guangdong, Zhejiang, Shandong, Jilin, and Beijing in China, a range of nutritional and health-care products are being developed from numerous artificially reared edible insect species. The edible insect products essentially include health beverage products like tonic and nutrition wine made from silkworm pupae, moths, fairy moths, and other products like Cordyceps essence and scorpion essence oral liquid, while oral liquid utilizing black ants is also immensely popular. These edible insect-based products have gained popularity among consumers and are even being exported overseas as a foreign income source. The practice of rearing silkworms for silk production and extracting pupal oil for consumption or the creation of health products and medicines has a long-standing tradition in China and TCM practices. Furthermore, China also incorporates pine caterpillar pupae oil into its dietary practices, suggesting an immense industrial prospect for the development of edible insect enterprises [22]. The extraction of oil from edible insects, known for their high-fat content and ease of rearing and cultivation, has gained attention. This oil can be utilized in various ways, such as health care products, medicinal purposes, or even as industrial oil. The continuous improvement and extensive utilization of insect oil extraction technology is of paramount importance in supporting the utilization of edible insects and promoting their industrial development [22].

#### 4.2.4. Utilization of Edible Insects in the Yunnan Province of China

Yunnan is a multi-ethnic province in China, and there is an abundance of species of edible insects in this region, while the utilization methods and species of edible insects are also rich and diverse among different ethnic groups [63,161,162,163]. Based on some typical local edible-insect cuisines of the Yunnan Province, the species diversity, special cuisines, edible stage, host plants and distribution regions of edible insects, as well as the ethnic minorities with the habit of eating insects were summarized and organized into the following Table 2.

The larvae of *Corydalus cornutus*, belonged to Megaloptera and named as hellgrammites, can be cooked as the “soft-fried hellgrammites” bythe Bouyei nationality; they can feed the insects of *Nilaparvata lugens*, *Odontota scapularisin* and *Aphidoidea* aphids in rice fields [164]. *C. cornutus* is widely distributed almost throughout Yunnan, and people mainly eat its adults and larvae [163]. In Hymenoptera, *Apis* sp. are generalist herbivores, while *Vespa* sp., *Polistes* sp. and *Provespa* sp. are predatory wasps on cotton plants, and the people of Dai and Zhuang minorities in the regions of Xishuangbanna and Dehong have various ways of cooking the pupae of *Apis* sp., *Vespa* sp., *Polistes* sp., and *Provespa* sp., including deep-fried pupae, pupa paste, Dai-taste pupae, and crunchy wasps [161]. The species of edible ants belonging to Orthoptera are more extensive, and they can host on host plants of *Dalbergia hupeana*, *Ficus carica* and *Vachellia farnesiana*. The people of Wa, the Miao minorities, and other ethnic groups in the regions of Lincang, Pu’er, and Xishuangbanna have the habit of eating ants, including the local delicacies of salad and pastry of ant eggs, and acid ant vinegar [165,166]. Gryllus bimaculatus mostly feed on the host plants of *Glycine max*, *Oryza sativa*, *Zea mays*, and *Setaria italica*, and *Locusta migratoria* and *Oxya chinensis* mostly feed on the host plants of *Triticum aestivum*, *O. sativa*, *Z. mays*, *Sorghum bicolor*, while the people of the Dai minority all over the Yunnan Province mainly eat the adults of *G*. *bimaculatus* (including cricket jam and fried cricket) and the people of Yi and Hani minorities all over the Yunnan Province eat the adults and larvae of *L*. *migratoria* and *O*. *chinensis* [167,168].

The people of the Wa minority in the regions of Xishuangbanna, Pu’er, Lincang, Dehong, Baoshan, and Wenshan eat the larvae of *Xylotrechus quadripes* belonging to Coleoptera by frying, and *X*. *quadripes* is also known as the “Chai worm” and its host plants are *Coffea arabica*, *Lannea A*., and *Tectona grandis* [169]. The people of the Dai minority in the east-south regions of the Yunnan Province eat the adults of *Brontispa longissima* by frying, and *B*. *longissima* is also known as “Coconut worm” and hosts on the plants of *Cocos nucifera*, *Phoenix canariensis*, *Washingtonia filifera* and *Phoenix roebelenii* [170]. *Aspongopus* sp., belonging to Hemipteranare and named as aspongopus, feeds on *Cucurbits* plants, while the people of Dai minority in the regions of Honghe, Baoshan, and Wenshan eat the adults of *Aspongopus* sp. by deep frying [171]. The people of the Dai minority in the regions of Xishuangbanna and Dehong minorities eat the nymphs of *Cryptotympana atrata*, and there are four types of cicada food (including fermented cicada, cicada pupa cake, cicadas chop raw and cicada meat balls). *C*. *atrata* belongs to Hemipteranare and mainly hosts on the plants of *Sophora japonica* and *Salix babylonica* [172]. The “fried granulation” is made by the larvae (named as maggots and scavengers) of *Musca domestica* belonging to Diptera; it is the local special insect food of the people of the Hani minority in the regions of Dehong and Xishuangbanna [173]. The edible insect forms of Lepidopteran are mainly larvae and pupae [163]. Among them, *Omphisa fuscidentalis* hosts on the *Bambusoideae* plants [174] and *Antherea pernyi* hosts on the plants of *Quercus* sp. [175]. The people of Dai, Hani, Jino, Zhuang and Buyi minorities in the regions of Xishuangbanna, Dehong, and Pu ‘er eat the larvae of *O*. *fuscidentalis* by frying [174], and those of the Yi minority in the regions of Qujing, Dehong, Zhaotong, and Pu ‘er eat the crispy silkworm pupae of *A*. *pernyi* [175]. The people of the Zhuang minority in the Yunnan Province cook and eat the larvae of *Clanis bilineata* (named as “spiced bean worms” and also known as DanBean), and *C*. *bilineata* mainly feed on the host plants of *Oryza sativa*, *Robinia pseudoacacia*, *Vigna radiata*, and *V*. *unguiculata* [176].

## 5. Production Advantages of Edible Insects over Traditional Livestock

### 5.1. Short Life Cycle

The average daily weight gain (ADWG) is an indicator of protein production efficiency. The ADWG of insects (including edible individuals) ranges from 4% to 19.6%, which is 25% higher than the ADWG of pigs and 600% higher than that of cattle [177,178]. In addition, insects rapidly grow to adulthood much faster than traditional livestock, and many species of insects can produce thousands of offspring, whereas a conventional livestock animal produces only a few offspring. The offspring of these insects reach adulthood in a matter of days, in contrast to the months or years required for poultry or ruminants. Therefore, insects (including edible insects) can produce protein at a much higher rate than traditional livestock [179].

### 5.2. Low Production Costs

Most edible insects are phytophagous and their food sources are very wide, easily available, and inexpensive [66]. Edible insects are relatively simple to feed with low input costs. Various types of plants, including wild and cultivated varieties, as well as humus and animal excreta, can be used as insect feed. On average, 1 kg biomass of insects can be raised on 3.2 kg of feed [71]. In 2015, the FAO Report highlighted that the environmental and economic costs of edible insects are significantly lower than those of livestock animals, both in terms of the greenhouse gas emissions per kg of live organisms and in terms of the land and water resources they require [20]. If edible insects are produced on a large scale, it will be less costly and more profitable than raising traditional livestock (e.g., cattle and pigs) [180].

### 5.3. Improvement of the Ecological Environment

Livestock production is one of the most ecologically harmful of all anthropogenic activities. It is evident that animal husbandry contributes significantly to global warming, in addition to causing extensive ecological degradation [179]. Livestock is a significant contributor to the emission of greenhouse gases with N_2_O, accounting for 65% of the global anthropogenic emission, and this represent 75% to 80% of all agricultural emissions, and it is projected to increase in the coming decades [181]. As a novel source of food, edible insects can be a viable alternative to mitigate environmental degradation through the reduction in greenhouse gases emission compared to conventional livestock [182]. Oonincx et al. (2010) reported that the greenhouse gas emissions from four of the five insect species they examined were considerably lower than those from pigs and ruminants. The CO_2_ production per kg of body mass gain (BMG) in insects (337 g/kg) is 39% of the CO_2_ production per kg of BMG in pigs and 12% of that in cattle [178]. Oonincx and de Boer (2012) also observed that insects released a quantity of ammonia gas (NH_3_) in the range of 3.0 to 5.4 mg BWG/kg feed per day [177]. The NH_3_ emission from pigs was equivalent to 812 times that of crickets and 50 times that of locusts [177,178].

### 5.4. Water Saving

The key variable measuring the water required to produce livestock is “feed conversion efficiency”, which refers to the amount of food required to produce a given amount of final product, such as meat, eggs, etc. [183]. The fewer greenhouse gases emitted by edible insect farming, the less water and space needed, which means lower economic investment with higher feed conversion efficiency [184]. The amount of water required to produce 1 kg animal protein of livestock is approximately 100 times greater than the amount of water required to produce 1 kg cereal protein [185]. It has been demonstrated that, for many crickets and mealworms, it requires only 40 L water to produce 1 kg of insect protein [179]. Pimentel (1997) estimated that 500 to 2000 L water was required to produce 1 kg potatoes, wheat, rice or soybeans, while approximately 43,000 L water would be required to produce 1 kg beef [186].

### 5.5. Reduced Land Use

As the global demand for meat-based food continues to grow, the pressure on the land resources required by farmers to raise livestock is increasing [187]. Oonincx and de Boer (2012) indicated that livestock occupied about 70% of the global available agricultural land, and that the land used to produce the mealworm protein per hectare was 2.5 hectares, 2–3.5 hectares, and 10 hectares needed to produce the same amount of milk, pork/chicken, and beef protein respectively [177]. It is not difficult to conclude that the land use required for insect protein production is significantly smaller than that required for traditional livestock.

## 6. Consumption of Edible Insects: Problems and Challenges

The popular proverb indicating that edible insects are the final gift from God to mankind highlights the crucial role these creatures play in the development of a prospective food sector [5]. The tradition of consuming edible insects is deeply rooted across different regions in China, and these edible insect dishes are savored by consumers with great delight. Nevertheless, the development of edible insects in China continues to encounter various obstacles.

### 6.1. Limited-Scale Production of Edible Insects

The large-scale production of edible insects faces a significant challenge due to the problems associated with artificial rearing, as well as the inadequate production and processing technologies of edible insect products. The successful rearing of edible insects requires the provision of specific and even additional protection measures, along with maintaining suitable temperature conditions throughout their entire life cycles [188]. However, the lack of well-developed production technology and insufficient financial support hinder the huge industrialization prospects of edible insects. This limitation not only impedes the industrialization of edible insects but also restricts the development of the edible insect product market in the long term [189].

### 6.2. Lack of Rearing Technology and Facilities

The industrialization of edible insects must adapt high-tech absolute artificial rearing technology and facilities in order to produce high-quality edible insects. Relying only on the natural resources alone for the development and utilization of edible insects will not suffice with the market demand. Therefore, more efforts should be made in regard to artificial rearing and industrial processing technology. Research teams from scientific institutes or universities should be encouraged to actively participate in joint public relations with enterprises so as to provide scientific and technological support for the rearing and processing of edible insects [190]. In addition, the domestic edible insect processing technology is still in the exploratory stage. Insect food processing technology market is dominated by prototype food (directly eaten), while the poor sensory experience of prototype insect food not only aggravates the fear of consumers but also limits the industrial development of edible insect food [20].

### 6.3. Immature Market and Lack of Industrial Alliance

The industrial development of edible insect products in China has already begun with the appearance of some feeding and processing factories, such as the “Insect Industry Information Center of Shandong Insect Industry Association” (raising yellow mealworms), “Kunwei Wild Food Trading Company, Ningbo, China” and “Shenyang Vitality Food Co., Ltd., Shenyang, China”. However, most edible insect product-processing enterprises do not have the consciousness of industrial cluster and do not know how to focus on the professional division of labor. They do not make full use of infrastructure and business information sharing. Numerous edible insect products production enterprises are operating in the Shandong Province, such as Weifang Chunyuan Insect Company (Weifang, China) and Shandong Insect Company (Yantai, China). These enterprises can aggravate to form an industrial alliance of edible insect production and expand the competitiveness of enterprises. Due to the lack of cluster awareness of these enterprises, the market development of edible insect products has been impeded [189].

### 6.4. Low Acceptability of Edible Insects and Their Products

Consumer antipathy has greatly restricted the popularity of edible insects owing to their uncomfortable prototype appearance. Hu et al. (2013) conducted an interview revealing that 98.7% of respondents had heard of edible insect products where 3.9% of respondents were very familiar with edible insect products, 46.1% of respondents had a slight understanding, and 35.7% of respondents were relatively well familiar with comprehensive knowledge [191]. In addition, the inadequate processing techniques involved in the edible insect food production process contribute to the prevalent insect prototype among the consumers, which aggravates the antipathy of consumers. Therefore, eliminating consumer’s antipathy by providing a feasible environment for choosing mesmerizing edible insect products is a necessary first step in developing the market of edible insect products. It is evident that there is a need for public consumers to enhance the promotion of edible insects and their products via consumer awareness to eliminate their antipathy to edible insects.

### 6.5. Safety of Consumption

Edible insects and their products have a very high nutritional and medicinal value, but enhancing consumer safety and removing the preconception and bias towards edible insects and their products requires some fundamental approaches to be addressed. For instance, some specific consumers may have allergies after consuming edible insects due to the presence of complex macromolecules like chitin and alkaloids. The chitin content in edible insects can reach 20~60%, so the development and utilization of edible insects should be marked with the relevant information on allergens. Moreover, Zhou et al. (2021) mentioned that some toxins might remain in edible insects, which occurs when insects are fed with moldy raw materials contaminated by mycotoxins (especially plant raw materials) [192]. Additionally, due to moisture exposure during storage and transportation and the packaging bags being contaminated by mycotoxins, there is a suitable environment for mold development, leading to decreased acceptability of edible insects and their products [192]. On the other hand, the residual heavy metals in soil and water will accumulate through the food chain of edible insects. Therefore, safe edible insect rearing requires a hygienic rearing environment and raw materials (including water sources) in order to reduce the heavy metals accumulation in edible insects [192]. In addition, edible insects also carry some pathogenic bacteria that can cause foodborne diseases, and some edible insects pose a potential threat of being a carrier of fatal pathogenic bacteria, which are the hidden dangers of microbial foodborne diseases [192]. Chemical pesticides can remain in the bodies or surfaces of edible insects and eventually cause damage to the human nervous and reproductive systems through the food chain [190]. Therefore, precautions must be followed during the feeding of edible insects when they are reared artificially in order to produce green, natural and organic edible insects which will be later employed for the industrial production of edible insect products.

### 6.6. Legislation on Edible Insects in China

In America, the Food and Drug Administration (FDA) supervises and manages insects as food based on the Federal Food, Drug, and Cosmetic Act. Edible insects can be recognized as safe if the company or a third party demonstrates a relevant scientific opinion based on published or unpublished scientific data [193]. In Europe, although there is currently no specific legislation regulating the consumption of edible insects, the European Commission maintains rigorous regulations concerning food safety and quality for edible insects. In May 2017, the Swiss government amended the food regulations to permit the use of grasshoppers, yellow mealworms, and crickets as ingredients in food products, and Switzerland became the first European country to openly sell insect-based food products [20]. In late 2017, the Finland government lifted its restrictions on insect-based foods, and in early 2018, the European Union (EU) officially added edible insects into the “Novel Food Catalogue” [20]. All these regulations extend to both imported and exported food products as well as the entirety of the EU’s food production and processing chain. The insect-derived food products are classified as novel foods and require authorization before they can be introduced to the EU market [194]. The enforcement of these regulations was carried out under the auspices of the new Regulation 2283/2015 on Novel Foods, along with its implementing Regulations 2468/2017 and 2469/2017 [195].

In China, edible insect consumption is prevalent, while there are no specific laws regulating the use of edible insects for human consumption [196]. The food catalogue lists only silkworms and yellow mealworms as food sources [20]. However, the sale and consumption of certain species of edible insects, especially those that are poisonous or endangered, are strictly prohibited by Chinese laws [196]. The situation not only in China but also in American, Europe, and other places regarding the edibility of insects is much more controversial than how it is presented as insect foods. And globally, the legislation of rules and regulations, both at the national and international level, on the production, storage, and consumption of edible insects as food/feed ingredients are often absent or, at best, non-exhaustive, which should be further explored and given more attention.

## 7. Developmental Prospect of Edible Insect Food

Though the concept of edible insect food enterprises emerged in the late 1980s, the development of edible insect foods accelerated at pace later when the debate of sustainable food measures to combat climate change and food insecurity started at the government policy level. The Review of Big Agriculture and Big Food is repeatedly mentioned in the annual No. 1 Document of the Central Government of China, and it is officially emphasized by integrating a wider array of protein sources derived from cultivated land, grassland, forests, and oceans, as well as plants, animals and microorganisms. Through decades of exploration and development, China’s edible insect industry has developed rapidly and achieved a remarkable scale. Millions of tons of quick-frozen Chinese rice locusts are exported to Japan each year, which promotes the development of China’s international trade. Additionally, currently there are many insect industries in many regions of China, such as “insect wine” in the Guangdong Province and “triloite tea” in the Hunan Province. In Yantai City of the Shandong Province, cicadas are edible insects with high acceptance and popular consumption in daily life, and the local production of “Fried cicada” is exported to Japan, with the price being as high as RMB 120,000 per ton [180]. From an economic point of view, the edible insect market is huge and full of unlimited business opportunities. However, much more financial investment should be invested in new processes of edible insect products and their factory production in order to achieve large-scale insect rearing and processing of industrial food [195]. Therefore, the development of the edible insect market and the expansion of the rearing scale have great potential benefits in regard to the processors and exporters obtaining the required quantity and quality of insect products [197]. Furthermore, it is essential to establish well-coordinated regulatory frameworks and digital platforms for exchanging information in order to systematically advance the development of edible insect-rearing enterprises. It is crucial for both the general public and policymakers to fully comprehend the extensive social, economic, and environmental advantages, as well as the potential impacts, associated with these edible insect enterprises on the circular economy. Edible insect producers should quickly obtain tailored feeding technologies and have full-fledged access to market information so that they can achieve edible insect scale rearing and expand the edible insect sales market [197].

The primary goal of the edible insect as an industrial enterprise is to replace traditional food proteins with edible insect food products as dietary supplements and alternative protein sources, thereby promoting cost-effective, affordable, and environmentally benign edible insect food products. So much more research on optimal processing methods is needed in order to achieve the best compromise of bio-active function, delicacy, cost-effectiveness, sustainability, and consumer safety regarding edible insect food through incorporating edible insect protein into the large-scale food industry and daily diets of consumers [18,198]. In addition, the allergens and desensitization technology of edible insects must be focused on via extensive research. Lastly, a focus should be placed on improving the laws and regulations related to edible insects to ensure consumers about well-developed processing mechanisms and consumer safety [199]. Edible insect protein is a vibrant nutrient source with medicinal properties and it has the attractive prospect of being an emerging source of alternative protein resources, attracting the interest of scientific researchers lately. Prolific focuses on enhancing artificial rearing and cultivation technologies to maximize the value of insect protein resources are underway. These efforts will ultimately promote the prosperity of the insect-rearing industry and mobilize a large number of people for idle labor, fostering a self-sustaining cycle for the edible insect trade [200]. The continuous advancements in edible insect food rearing and the development of sophisticated and well-processed edible insect products will undoubtedly play a significant role in diversifying public food options and ensuring a reliable and sustainable food supply.

## Figures and Tables

**Figure 1 foods-13-01986-f001:**
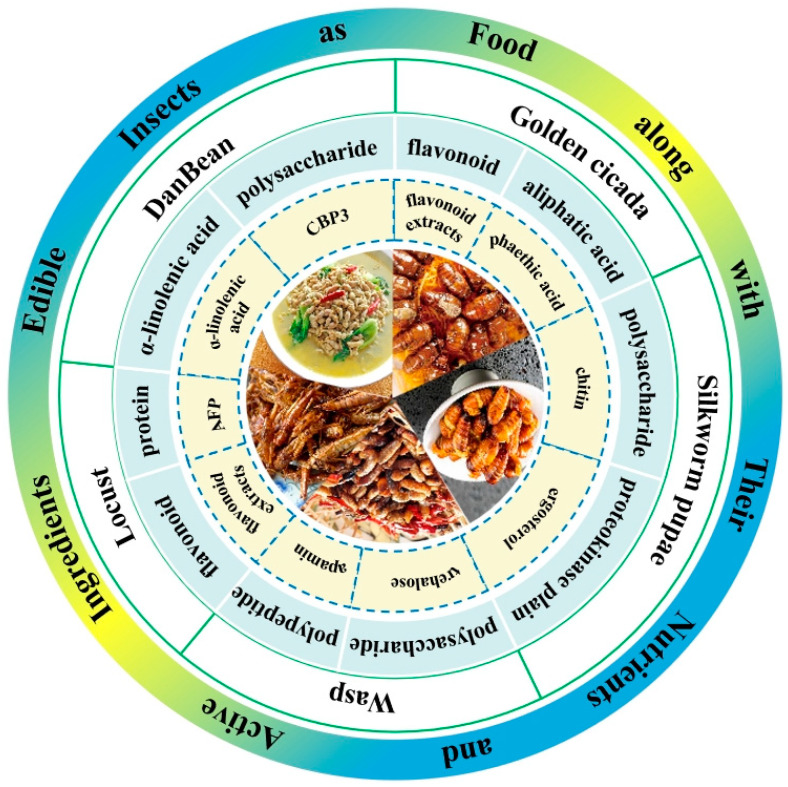
Five respective species of edible insects (including golden cicada, DanBean (i.e., the larvae of soybean hawk moth *Clanis bilineata tsingtauica*), silkworm pupae, wasps and locusts) as food along with their nutrients and active ingredients (**Note:** CBP3—*C. bilineata tsingtauica* polysaccharides; AFP—antifreeze proteins).

**Figure 2 foods-13-01986-f002:**
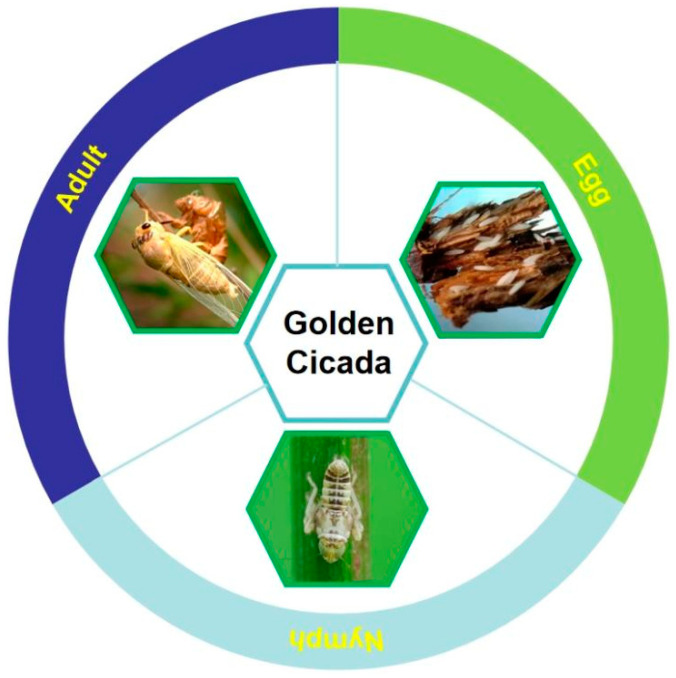
Life cycle of golden cicada *Crytotympana pustulata* (**Note:**
*C. pustulata* has three distinct developmental stages of egg, nymph, and adult; nymphs and adults are mainly the edible ones).

**Figure 3 foods-13-01986-f003:**
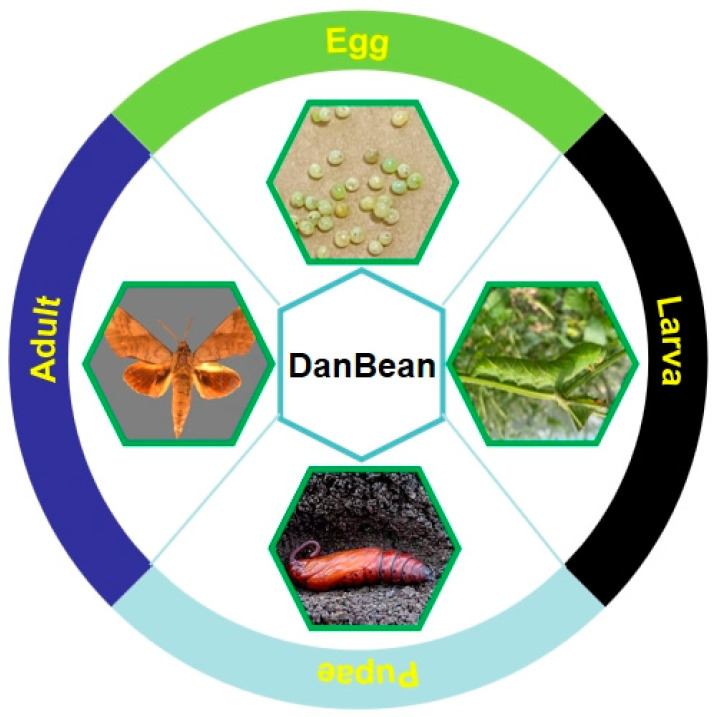
Life cycle of the soybean hawk moth *Clanis bilineata tingtauica* (**Note:**
*C. bilineata tingtauica* has four distinct developmental stages of egg, larva, pupa and adult; larvae are the edible ones, named as DanBean).

**Figure 4 foods-13-01986-f004:**
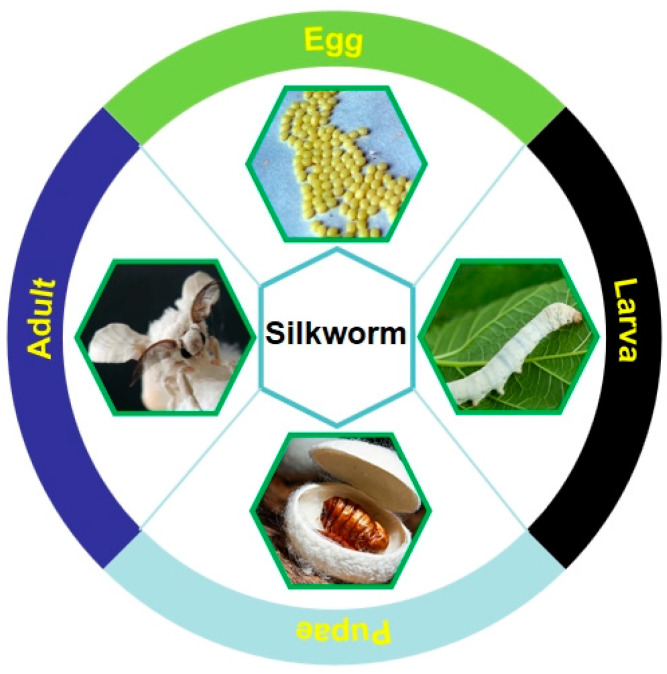
Life cycle of the silkworms (**Note:** Silkworms have four distinct developmental stages of egg, larva, pupa and adult; pupae are the edible ones).

**Figure 5 foods-13-01986-f005:**
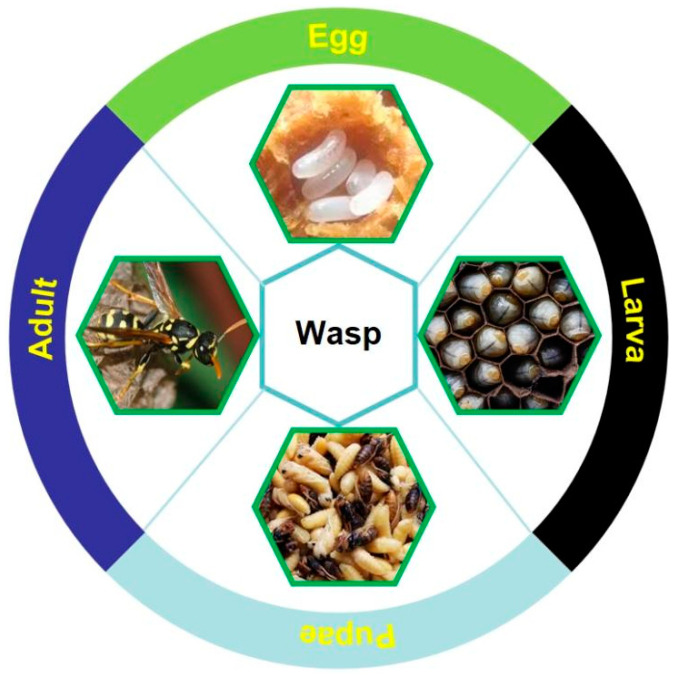
Life cycle of the wasps (**Note:** Wasps have four distinct developmental stages of egg, larva, pupa and adult; adults are the edible ones).

**Figure 6 foods-13-01986-f006:**
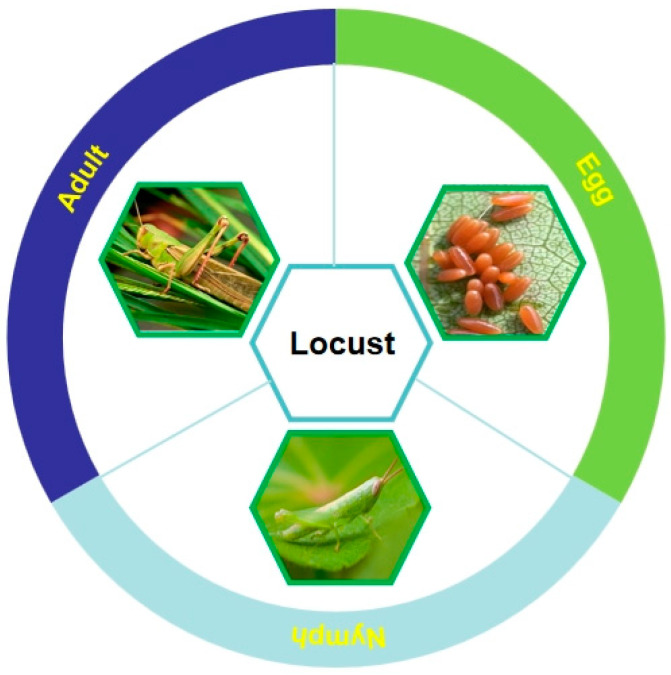
Life cycle of the locusts (**Note:** Locusts have three distinct developmental stages of egg, nymph and adult; nymphs and adults are the edible ones).

**Table 1 foods-13-01986-t001:** Special bio-active substances in edible insects.

Scientific Name	Compounds	Type	Function Values	References
*Isopterans*	Interferon	Protein	Inhibit tumor formation	[149]
*Bombyx mori*	Pupal protein	Protein	Reduce the risk of cardiovascular diseases and cancer	[126]
*Ericerus pela*	Phosphatide	Lipid	Invigorate brain functions, lower blood lipids, cholesterol removal, treatment of fatty liver, cirrhosis and anti-aging.	[132,151]
*Apis mellifera carnica*	Trehalose	Polysaccharide	Enhance body immunity and anti-radiation	[46]
*Dendrolimus houi*	Chitin	Polysaccharide	Health care functions due to dietary fiber and anti-thrombotic	[17,151]
*Hepiaua larva* and *Bombyx mori*	Cordyceps polysaccharide	Polysaccharide	Antioxidant and immune regulation, anti-tumor activities, protect kidney and liver, reduce blood glucose and lipids, anti-radiation, anti-aging.	[143,144]
*Hyalophora cecropia* and *Sarcophaga percgrina*	Antibacterial peptide	Peptide	Inhibit tumor formation	[150]
*Gryllodes sigillatus*	Anti-inflammatory peptide	Peptide	Angiotensin-converting enzyme inhibitor	[153]
*Oxya chinensis*	Flavonoid extracts	Flavonoid	Hypolipidemic, anti-fatigue and antioxidation functions	[152]

**Table 2 foods-13-01986-t002:** General situation of the edible insect resources in Yunnan Province of China.

Insect Order	Scientific Name	Special Cuisine	Edible Stage	Ethnic Minority	Host Plants	Distribution Regions	References
Megaloptera	*Corydalus cornutus*	Soft-fried hellgrammites	Adult, larva	Bouyei	Rice (*Oryza sativa*) plants (Megalopterans larvae feed on *Nilaparvata lugens*, *Aphidoidea*, *Odontota scapularis*)	All over Yunnan	[164]
Hymenoptera	*Apis cerana*, *Vespa* sp., *Polistes* sp., *Provespa* sp.	Deep-fried pupae	Larva, pupa	Basically all	Cotton (*Gossypium hirsutum*) plants (*Apis* sp. are generalist herbivores while *Vespa* sp., *Polistes* sp. and *Provespa* sp. are predatory wasps)	Xishuangbanna, Dehong	[161]
		Pupa paste		Dai			
		Dai-taste pupae					
		Crunchy wasps		Zhuang			
Orthoptera	*Oecophylla* sp.	Ant eggs salad	Spawn	Dai	*Dalbergia hupeana*, *Ficus carica*, *Vachellia farnesiana*	Lincang, Pu ‘er, Xishuangbanna	[165,166]
		Ant egg pastry		Va			
		Fried flying ants		Miao			
		Acid ant vinegar	Larva	Dai, Jingpo, Miao, Jinuo			
	*Gryllus bimaculatus*	Cricket jam	Adult	Dai	*Glycine max*, *Oryza sativaZea mays*, *Setaria italica*	All over Yunnan	[167]
		Fried cricket					
	*Locusta migratoria*, *Oxya chinensis*	Fried locust	Adult, larva	Yi	*Triticum aestivum*, *Oryza sativa*, *Zea mays*, *Sorghum bicolor*	All over Yunnan	[168]
		Grasshopper jam		Hani			
Coleoptera	*Xylotrechus quadripes*	Fried bamboo worm	Larva	Va	*Coffea arabica*, *Lannea A.*, *Tectona grandis*	Xishuangbanna, Pu’er, Lincang, Dehong, Baoshan, Wenshan	[169]
	*Brontispa longissima*	Fried coconut worm	Adult	Dai	*Cocos nucifera*, *Phoenix canariensis*, *Washingtonia filifera*, *Phoenix roebelenii*	East-south regions	[170]
Hemiptera	*Aspongopus* sp.	Deep-fried aspongopus	Adult	Dai	*Cucurbits*	Honghe, Baoshan, Wenshan	[171]
	*Cryptotympana atrata*	Fermented cicada	Nymph	Dai	*Sophora japonica*, *Salix babylonica*	Xishuangbanna, Dehong	[172]
		Cicada pupa cake					
		Cicadas chop raw					
		Cicada meat balls					
Diptera	*Musca domestica*	Fried granulation	Larva	Hani	Doesn’t have a specific host plants as it is a scavenger	Dehong, Xishuangbanna basin valley	[173]
Lepidoptera	*Omphisa fuscidentalis*	Fried bamboo worm	Larva	Dai, Hani, Jino, Zhuang, Buyi	*Bambusoideae*	Xishuangbanna, Dehong, Pu ‘er	[174]
	*Antherea pernyi*	Crispy silkworm pupa	Pupa	Yi	*Quercus* sp.	Qujing, Dehong, Zhaotong, Pu ‘er	[175]
	*Clanis bilineata*	Spiced bean worm	Larva	Zhuang	*Oryza sativa*, *Vigna radiata*, *Vigna unguiculata*, *Robinia pseudoacacia*	All over Yunnan	[176]

## Data Availability

No new data were created or analyzed in this study. Data sharing is not applicable to this article.

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
