# Peer review of "Research Progress and Production Status of Edible Insects as Food in China"

_foods, 2024, doi:10.3390/foods13131986_

Round 1
Reviewer 1 Report
Comments and Suggestions for Authors
In my opinion, the authors have made a significant contribution to the scientific community's understanding of this subject. The review manuscript is well-written and offers valuable information. However, I would like to provide some comments that I believe would further enhance the manuscript's quality before possible consideration for publication in the Journal as detailed in the following:
Manuscript ID: foods-3019740
Title: Research Progress and Production Status of Edible Insects as Food in China
This review provides that edible bugs are a beneficial lasting likewise greatly remote resource of food as well as medication. Additionally reveals the raised growth together with use of edible pests as an important food source with a substantial influence on sustainability and also human health and wellness. The authors highlight the list below finding:
1- Edible pests are abundant in healthy protein essential amino acids, healthy and balanced fats, minerals (specifically zinc as well as iron), vitamins, along with various other bioactive substances. They are a great resource of healthy protein, exceeding conventional animals’ resources like beef, pork, as well as hen in many cases.
2- Many edible bugs and also their byproducts have different medicinal residential properties consisting of anti-inflammatory, antioxidant antihypertensive antidiabetic, as well as anticancer impacts. Some, like Cordyceps sinensis, are very valued in conventional Chinese medication.
3- Edible bugs use countless benefits for food manufacturing, consisting of rapid recreation prices, reduced feeding prices, as well as marginal ecological effect. Their usage can add to lasting farming along with battle food instability.
4- The approaches of including edible bugs right into the diet plan varying from straight usage to refining right into different food like treats, drinks and also health and wellness supplements.
5- The evaluation discusses the quality, conservation, as well as medical worth of edible pests. It's vital to keep in mind that the lasting conservation of edible pests is crucial for their assimilation right into the food supply chain.
6- The difficulties such as minimal manufacturing range, absence of innovation, customer antipathy plus safety and security worries. They recommend options via research study, plan advancement, and also market partnership.
In my opinion, the authors have made a significant contribution to the scientific community's understanding of this subject. The review manuscript is well-written and offers valuable information. However, I would like to provide some comments that I believe would further enhance the manuscript's quality before possible consideration for publication in the Journal as detailed in the following:
- Abstract needs more revision to be attractive for the reader.
- There are minor errors in the spelling, grammar, and punctuation marks throughout the MS. Kindly revise it carefully.
- Revise format of all references, check the journal requirements and maintain consistency.
- Added section about the nutritional benefits of edible insects compared to traditional livestock? The abstract mentions that edible insects have high nutritional value, including protein and functional values. It would be interesting to explore the specific nutritional benefits of edible insects and how they compare to traditional livestock.
- It would be useful to examine the ways in which edible insects can contribute to food security, including their availability, affordability, and nutritional value.
- What are the environmental benefits of edible insect production compared to traditional livestock farming?
- What are the challenges to the adoption of edible insects as a food source, particularly in Western countries?
- How can edible insects be farmed and produced sustainably and at scale?
- What are the food safety concerns associated with edible insects, and how can they be addressed?
- How can edible insects be integrated into traditional food systems and cuisines?
- What is the current state of legislation and policy regarding edible insects as a food source, and how can it be improved?
- What are the potential health benefits and risks associated with consuming edible insects?
- How can edible insects be promoted and marketed to consumers, particularly in Western countries?
Author Response
Dear Reviewer,
How are you! Thanks for your comments for us to improve our review manuscript, the following is our responses one by one to your queries (Q), please check our responses, and the revised version of foods-3019740-R1.
The queries, and our responses were following as:
Q1: Abstract needs more revision to be attractive for the reader.
Response: Thanks! We have revised the Abstract of the version of foods-3019740-R1.
Q2: There are minor errors in the spelling, grammar, and punctuation marks throughout the MS. Kindly revise it carefully.
Response: Thanks! We have re-corrected and revised the spelling, grammar and punctuation marks in the version of foods-3019740-R1, please check the traced version of foods-3019740-R1.
Q3: Revise format of all references, check the journal requirements and maintain consistency.
Response: Thanks! We have revised all the format of cited references based on the journal requirements. Please check the version of foods-3019740-R1.
Q4: Added section about the nutritional benefits of edible insects compared to traditional livestock? The abstract mentions that edible insects have high nutritional value, including protein and functional values. It would be interesting to explore the specific nutritional benefits of edible insects and how they compare to traditional livestock. It would be useful to examine the ways in which edible insects can contribute to food security, including their availability, affordability, and nutritional value.
Response: Thanks for your good suggestion. We added one new Chapter 5 named as “Production advantages of edible insects over traditional livestock”. About the nutritional benefits of edible insects compared to traditional livestock, which were introduced in the respectively sections of nutrient values of edible insects in the version of foods-3019740-R1, e.g., 2.1.2-2.5.2, and the whole Chapter 3.
Q5: What are the environmental benefits of edible insect production compared to traditional livestock farming?
Response: Thanks for your good suggestion. We added one new Chapter 5 named as “Production advantages of edible insects over traditional livestock”. Besides 5.3 Improvement of the ecological environment, “5.1 Short life cycle”, “5.2 Low production costs”, “5.4 Water saving” and “5.5 Reduced land use” were also introduced compared to traditional livestock farming in the version of foods-3019740-R1.
Q6: What are the challenges to the adoption of edible insects as a food source, particularly in Western countries? How can edible insects be farmed and produced sustainably and at scale? What are the food safety concerns associated with edible insects, and how can they be addressed? How can edible insects be integrated into traditional food systems and cuisines? What is the current state of legislation and policy regarding edible insects as a food source, and how can it be improved? What are the potential health benefits and risks associated with consuming edible insects? How can edible insects be promoted and marketed to consumers, particularly in Western countries?
Response: Many thanks for your important meaningful questions about edible insects, all these are we want to treat now and in future! About the challenges to the adoption of edible insects as a food source not only in China but also in Western countries, the faced challenge is the positive perception and cognition of the public to edible insects, firstly to make people knowing and accepting edible insects as food even they are bad even awful appearance or terrible odor etc. So this review is necessary for the public to know and recognize edible insects. And the legislation on edible insects in China was introduced compared with those in American and Europe, seen in Line 40 of Page 30 – Line 20 of Page 31 in the version of foods-3019740-R1.
Reviewer 2 Report
Comments and Suggestions for Authors
The reviewed work is extremely extensive, well planned and written. I believe that after taking into account the following minor comments, it is suitable for publication.
Figure 1Informations written in the inner green circle of this chart are illegible.
Chapter 4 seems completely unnecessary. Bees are not treated as edible insects and their products, although processed by insects, are products of plant origin (with the exception of royal jelly or venom). Other insect products can be discussed when discussing individual insects.
There is no reference, although brief, to the economic aspect of edible insect production in China in relation to the costs of other animal production, taking into account the prices of raw materials, labor and production profitability.
Author Response
Dear Reviewer, How are you!
Thanks for your comments for us to improve our review manuscript, the following is our responses one by one to your queries (Q), please check our responses, and the revised version of foods-3019740-R1.
The queries, and our responses were following as:
Comments: The reviewed work is extremely extensive, well planned and written. I believe that after taking into account the following minor comments, it is suitable for publication.
Response: Thanks for your positive opinion on out review.
Q1: Figure 1 Information written in the inner green circle of this chart are illegible.
Response: Thanks! We have re-made this figure to show the clear chart. In order to see more clearly the active ingredients of the five insects in the inner circle, we changed the green circle to a more distinct yellow circle and enlarged the font size. And based on the comments of the Academic Editor Comments for Author that “This appears to be a very thorough review. It would be improved by having a picture of each of the 5 insects that are discussed at length because they may not be familiar to an international readership.”, we made five figures (i.e., Fig.2 - Fig.6) respectively to show the main five species of edible insects in China in this version of foods-3019740-R1.
Q2: Chapter 4 seems completely unnecessary. Bees are not treated as edible insects and their products, although processed by insects, are products of plant origin (with the exception of royal jelly or venom). Other insect products can be discussed when discussing individual insects.
Response: Thanks for your comments! Based on yours and the comment of Reviewer 3 that “I believe that point 4, “Insect products as food” is completely off-topic”, we deleted the primary Chapter 4 from the version of foods-3019740-R1, and moved the sections of royal jelly, bee venom and silkworm silk to the Chapter 2 as “2.6 Insect products used as edible insects” in Line 40 of Page 18 – Line 22 of Page 19 in the version of foods-3019740-R1.
Q3: There is no reference, although brief, to the economic aspect of edible insect production in China in relation to the costs of other animal production, taking into account the prices of raw materials, labor and production profitability.
Response: Thanks for your comments. Based on yours and the comment of Reviewer 3 that “Added section about the nutritional benefits of edible insects compared to traditional livestock?”, we added Chapter 5 as “5. Production advantages of edible insects over traditional livestock” in Line 1 of Page 28 – Line 10 of Page 29 in the version of foods-3019740-R1.
Reviewer 3 Report
Comments and Suggestions for Authors
The work in general is interesting because the focus on edible insects is currently of great interest. However, several critical points emerge.
First of all, regardless of the quality of the work, since line numbers are not present in the PDF, it is very difficult to review it specifically. However, I believe that before a detailed review, the work needs restructuring. Below are some suggestions:
In the introduction, the authors discuss insects in general and then only dedicate space to the species golden cicadas (nymphs and adults), DanBean (Clanis bilineata tsingtauica larvae), silkworm pupae, wasps, and locusts, which are the major edible insects in China. The authors justify this choice because these are the most commonly consumed species in China. However, it is my opinion that in a work aiming to provide an overview of edible insects, the discussion should also include cricket (A. domesticus) and mealworm (T. molitor), as they are among the most studied species to date due to their high nutritional value. For example, the species of the order Orthoptera, including crickets, grasshoppers, and locusts, have an average protein content of 61% with variations ranging from 6 to 77%.
In the introduction, where the text reads: "Conversely, Europeans preferred to eat locusts, beetles, and ants and among Europeans Swedish was the heart of edible insect-consuming culture [10]," I find it to be simplistic. The situation in Europe regarding the edibility of insects is much more controversial than how it is presented, and globally, rules and regulations, both at the national and international level, on the production, storage, and consumption of insects as food/feed ingredients are often absent or, at best, non-exhaustive. I believe this aspect should be further explored.
Another aspect that should be improved is the structure of the text. Often, the reading isn't linear. For example, within the paragraph 2.1.1 History and status of golden cicada consumption, there is a discussion of nutritional values. Then, the authors talk about specific species and later return to general information about the main nutrients and active ingredients of edible insects. The text is confusing.
I believe that point 4, "Insect products as food," is completely off-topic.
Point 5.2 Utilization methods of edible insects does not discuss the use of insects as food additives. Currently, research trends and food innovations focus on food fortification with alternative, sustainable, and functional foods to improve the nutritional value of food by correcting nutrient or mineral deficiencies or enhancing health-promoting properties. In this context, insects represent one of the most valuable functional ingredients and interesting solutions for the food industry. I believe this is a fundamental point.
In general, I believe too much space is dedicated to irrelevant topics (see point 4 on Insect products as food), although important and interesting, they do not integrate into or add information to the context indicated in the abstract, which recalls the following aspects: 'This review provides an introduction to three key aspects of edible insects as food: freshness, long-term preservation, and medicinal value.' Furthermore, other aspects indicated as topics are not sufficiently addressed. The review also encompasses rearing and producing technologies, resource utilization, and industrial development in China.
In my opinion, I believe the paper requires a total restructuring, thus I propose a major revision
Author Response
Dear Reviewer,
How are you! Thanks for your comments for us to improve our review manuscript, the following is our responses one by one to your queries (Q), please check our responses, and the revised version of foods-3019740-R1.
The queries, and our responses were following as:
Q1: First of all, regardless of the quality of the work, since line numbers are not present in the PDF, it is very difficult to review it specifically.
Response: Thanks! It is our mistake not to give line numbers of the MS PDF file. In the version of foods-3019740-R1, we added the line numbers.
Q2: However, I believe that before a detailed review, the work needs restructuring.
Response: Thanks for your suggestion. We restructured this version of foods-3019740-R1, deleted the primary Chapter 4 “Insect products as food”, added one new chapter “5. Production advantages of edible insects over traditional livestock”, and added new sections of “2.6 Insect products used as edible insects” and “6.6 Legislation on edible insects in China”, and briefly introduced the edible insects (especially crickets and mealworms) consumed in American and Europe in Line 17-25 of Page 3 in the version of foods-3019740-R1.
Q3: In the introduction, the authors discuss insects in general and then only dedicate space to the species golden cicadas (nymphs and adults), DanBean (Clanis bilineata tsingtauica larvae), silkworm pupae, wasps, and locusts, which are the major edible insects in China. The authors justify this choice because these are the most commonly consumed species in China. However, it is my opinion that in a work aiming to provide an overview of edible insects, the discussion should also include cricket (A. domesticus) and mealworm (T. molitor), as they are among the most studied species to date due to their high nutritional value. For example, the species of the order Orthoptera, including crickets, grasshoppers, and locusts, have an average protein content of 61% with variations ranging from 6 to 77%.
Response: In China, these five species of edible insects, i.e., golden cicadas, DanBean (C. bilineata tsingtauica larvae), silkworm pupae, wasps and locusts, are the common consumed edible insects, which are directly consumed and eaten as food. Crickets and mealworms are commonly used as edible insects mainly as insect protein food and feed. In the version, we also added the introduction of crickets and yellow mealworms consumed in American and Europe in Line 17-25 of Page 3 in the version of foods-3019740-R1.
Q4: In the introduction, where the text reads: "Conversely, Europeans preferred to eat locusts, beetles, and ants and among Europeans Swedish was the heart of edible insect-consuming culture [10]," I find it to be simplistic. The situation in Europe regarding the edibility of insects is much more controversial than how it is presented, and globally, rules and regulations, both at the national and international level, on the production, storage, and consumption of insects as food/feed ingredients are often absent or, at best, non-exhaustive. I believe this aspect should be further explored.
Response: Many thanks for your comments and suggests. In our review, we want to pay more attention on the edible insects consumed in China, and the status of edible insects consumed in Europe and American were briefly introduced in the Introduction and further introduced in the first paragraph of “2. Major edible insect species and their resource utilization in China”. Indeed, globally, rules and regulations, both at the national and international level, on the production, storage, and consumption of insects as food/feed ingredients are often absent or, at best, non-exhaustive. So we added one section of “6.6 Legislation on edible insects in China”, the legislation on edible insects in China was introduced compared with those in American and Europe, seen in Line 39 of Page 30 - Line 20 of Page 31 in the version of foods-3019740-R1.
Q5: Another aspect that should be improved is the structure of the text. Often, the reading isn't linear. For example, within the paragraph 2.1.1 History and status of golden cicada consumption, there is a discussion of nutritional values. Then, the authors talk about specific species and later return to general information about the main nutrients and active ingredients of edible insects. The text is confusing.
Response: Thanks for your comments. We revised this paragraph, and deleted the information that “Live golden cicada nymphs are comprised of 58.58% protein, 10.23% fat, 0.58% total phosphorus, and 16.0mg/kg calcium and 82.2mg/kg Zinc (Zhu, 2011).”, and moved this sentence to Line 8 - 10 of Page 23 in the version of foods-3019740-R1.
Q6: I believe that point 4, "Insect products as food," is completely off-topic.
Response: Thanks for your comments! Based on yours and the comment of Reviewer 1, we deleted the primary Chapter 4 from the version of foods-3019740-R1, and removed the sections of royal jelly, bee venom and silkworm silk to the Chapter 2 as “2.6 Insect products used as edible insects” in Line 40 of Page 18 - Line 22 of Page 19 in the version of foods-3019740-R1.
Q7: Point 5.2 Utilization methods of edible insects does not discuss the use of insects as food additives. Currently, research trends and food innovations focus on food fortification with alternative, sustainable, and functional foods to improve the nutritional value of food by correcting nutrient or mineral deficiencies or enhancing health-promoting properties. In this context, insects represent one of the most valuable functional ingredients and interesting solutions for the food industry. I believe this is a fundamental point. In general, I believe too much space is dedicated to irrelevant topics (see point 4 on Insect products as food), although important and interesting, they do not integrate into or add information to the context indicated in the abstract, which recalls the following aspects: 'This review provides an introduction to three key aspects of edible insects as food: freshness, long-term preservation, and medicinal value. Furthermore, other aspects indicated as topics are not sufficiently addressed. The review also encompasses rearing and producing technologies, resource utilization, and industrial development in China. In my opinion, I believe the paper requires a total restructuring, thus I propose a major revision.
Response: Thanks for your comments. Indeed, it is one hot topic that the use of insects as food additives, and “edible insects used as food additives” is one interesting and novel topic and issues for us to prepare another review in future. In this Review, we focused on the aspects of edible insects as food: freshness, long-term preservation, nutritional and medicinal values. About the “point 4” (Insect products as food), we deleted from the version of foods-3019740-R1. Thanks again! And we have restructured the version of foods-3019740-R1 based yours and other two reviewers’ comments and suggestions.
Round 2
Reviewer 1 Report
Comments and Suggestions for Authors
The authors done all the required changes as mentioned in the comments.
In my opinion its suitable for publication in this form.